# Self-clocking fast and variation tolerant true random number generator based on a stochastic mott memristor

Gwangmin Kim [1], Jae Hyun In [1], Young Seok Kim [1], Hakseung Rhee[1], Woojoon Park[1], Hanchan Song [1], Juseong Park [1] & Kyung Min Kim [1✉]

The intrinsic stochasticity of the memristor can be used to generate true random numbers, essential for non-decryptable hardware-based security devices. Here, we propose a novel and advanced method to generate true random numbers utilizing the stochastic oscillation behavior of a $NbO_x$ mott memristor, exhibiting self-clocking, fast and variation tolerant characteristics. The random number generation rate of the device can be at least $40\ kb\ s^{-1}$, which is the fastest record compared with previous volatile memristor-based TRNG devices. Also, its dimensionless operating principle provides high tolerance against both ambient temperature variation and device-to-device variation, enabling robust security hardware applicable in harsh environments.

[1] Department of Materials Science and Engineering, Korea Advanced Institute of Science and Technology (KAIST), Daejeon, Republic of Korea. ✉email: km.kim@kaist.ac.kr

In future electronics, as the internet of things (IoT) expands the reach of technologies, chip authentication for secure communication will be increasingly important[1]. Accordingly, developing a hardware-based true random number generator (TRNG) is crucial, since it is intrinsically non-decryptable, unlike the software-based TRNG. For better security, researchers have proposed novel methods for realizing a TRNG, utilizing the stochastic switching devices such as spin transfer-torque, magnetic memory, and resistive memory[2–15]. The early TRNGs utilized the stochastic characteristics in the switching voltages or the programmed states of the non-volatile memristors. Although the proposed methods were viable, they commonly relied on stochastic variation in relation to reference values, which can be vulnerable if the reference value changes over time or over cycling.

To overcome the problem, H. Jiang et al. proposed a volatile memristor-based TRNG[6]. They observed the stochasticity in the switching delay in the Ag:SiO$_2$-based diffusive memristor and designed a novel circuit that converted the stochasticity to randomness by introducing a clock and counter. The device counted the number of clocks after the stochastic turn-on switching and digitalized it. This method suggested a way to generate dimensionless values that required no reference value to define the randomness. Afterward, K. S. Woo et al. introduced a nonlinear feedback shift register (NFSG) with a HfO$_2$-based volatile memristor to further increase the TRNG rate[8]. The NFSG can increase the bit size proportional to the number (n) attached to the device, making the rate n times faster. However, in both methods, the switching delay time was intrinsically long, several hundreds of μs, preventing fast TRNG operation. Also, they required an exterior clocking circuit, which can be difficult to integrate.

A NbO$_x$-based mott memristor shows a stochastic (or chaotic in specific condition) threshold switching behavior. Various applications have been proposed to exploit it such as the Hopfield network or fundamental oscillatory computing[16,17]. Considering such high usefulness of the NbO$_2$ mott memristor for various applications, its application to the TRNG should be reasonable, but no studies have demonstrated it yet.

In this study, we propose a self-clocking TRNG device utilizing the oscillating behavior of a mott memristor. We identified the origin of the randomness to be thermal fluctuations during oscillation, using a numerical simulation and a thermodynamic simulation. The oscillation speed is fast[18], allowing the fastest random bit generation rate of 40 kb s$^{-1}$ at least. Also, it has an inherent self-clocking capability, so that no external clocking circuit is needed. From the integrated device, we collected 130 million random bits and they passed all of the NIST random number tests successfully. Furthermore, we confirmed that the device worked well under a wide range of ambient temperatures, ranging from 300 K to 390 K, and in different characteristic devices, suggesting the high tolerance of the TRNG system.

## Results and discussion
### Stochastic oscillation of NbO$_x$ threshold switching device.
Figure 1a shows a top view optical microscopy image of a 5 μm × 5 μm cross-point NbO$_x$-based threshold switching (TS) device. The inset shows the device stack. The detailed device fabrication process can be found in the Methods section. Figure 1b shows the current-controlled negative differential resistance (NDR) characteristic of the device. An electroforming process was preceded by applying −5 V with a 5 mA of compliance current to the top Pt electrode. The device has two distinct NDR regions, at a low current ranging from 50 μA to 100 μA, and at a high current ranging from 400 μA to 550 μA. We designated them NDR-1 and NDR-2, respectively. Both NDRs are well understood as thermally

activated transport and mott transition, respectively[16,19–24]. NDR-1 is attributed to the thermally activated transport mechanism; as the current increases, the device temperature increases by Joule-heating, and this activates the carrier injection, leading to the current increase. Such positive feedback of carrier generation and Joule heating drastically reduces the device resistance to the NDR-1. When the temperature reaches around 1070 K at the higher current, mott transition occurs, producing the box-shaped NDR-2[16,21,22,25]. The inset in Fig. 1b shows the voltage-controlled threshold switching characteristic of the device, with a compliance current ranging from 0.1 mA to 1 mA for reference.

When a constant voltage is applied with a proper series resistor, the NbO$_x$ TS device can exhibit oscillating current behavior[16,26–29]. Figure 1c shows a set-up to test the oscillation characteristic of the device. In this configuration, when the applied bias exceeded the threshold voltage ($V_{th}$) of the device, the device turned on, and the device's potential decreased due to the voltage divider effect between the resistor and the on state of the device. Then, the TS device turned off spontaneously, resulting in the oscillation.

During the oscillation, the capacitive components in the device influence the frequency of the oscillation by inducing an RC delay, which is dependent on temperatures. Thus, the heat generation and dissipation conditions are stochastic, resulting in a slight variation in the oscillation period. Figure 1d shows current oscillation waves generated from the circuit. The pulse amplitude and width were 1.65 V and 34 μs, respectively, with a 1.8 μs of pulse rising and falling time. The load resistance was 2 kΩ. This included 20 waves, which were consecutively collected under the same conditions. The bottom left inset in Fig. 1d shows that all waves overlapped with a little variation at the beginning of the oscillation. However, as the stochastic oscillation repeated, as shown in the bottom right inset in Fig. 1d, their variation became irregular due to the accumulation of the variation, making the oscillation hard to predict.

Figure 1e shows the distribution of peak intervals ($t_{intvl}$) during the oscillations collected from 100 waves. The first and last distributions are shown as red and blue curves representatively, while others are in gray. The inset plots the average time of intervals and its standard deviation at each wave. It shows that both the average and deviation were fluctuating inconsistently, which is regarded as a completely random process.

### Thermal fluctuation induced stochastic oscillation modeling.
We conducted a numerical simulation to confirm the influence of the random thermal fluctuation on the stochastic oscillation. Because the threshold switching originates from a thermally activated transport mechanism, we adopted a 3D-modified Poole–Frenkel transport model to emulate the threshold switching characteristic of the device[16,20–22]. The following equation describes the model.

$$i_m = A V_m \left[ \sigma 0 e^{-\frac{E_a}{k_b T}} \left\{ \left(\frac{k_b T}{\beta}\right)^2 \left(1 + \left(\frac{\beta\sqrt{V_m/d}}{k_b T} - 1\right) e^{\frac{\beta\sqrt{V_m/d}}{k_b T}}\right) + \frac{1}{2d} \right\} \right],$$

(1)

where $k_b$ is the Boltzmann constant, $d$ and $A$ are the thickness and area of the switching region, respectively. The $\sigma_0$, $\beta$, and $E_a$ are material constants[20,30]. To represent the mott transition, we introduced an abrupt change in thermal resistance from $1.8 \times 10^6$ KW$^{-1}$ to $2.1 \times 10^6$ KW$^{-1}$ at 1070 K, the mott transition temperature of NbO$_2$. Considering the thermodynamics, we adopted Newton's cooling law with a harmonic thermal fluctuation, as

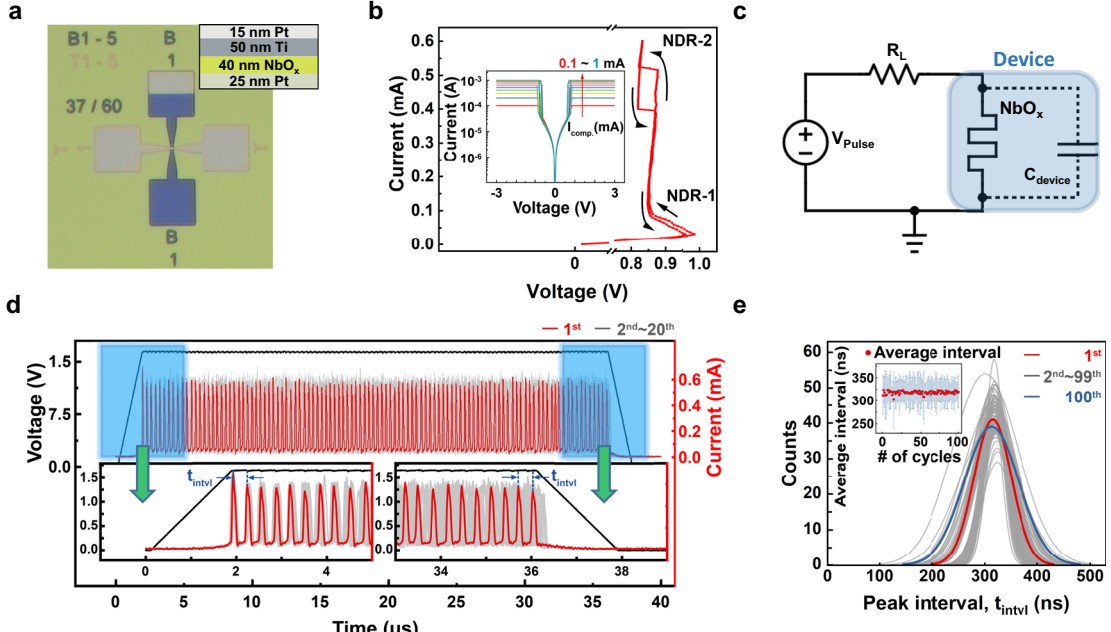

**Fig. 1 Electrical properties of NbOx-based threshold switching (TS) device, and its stochastic oscillation. a** An optical microscope image of the NbOx-based two terminal device. The inset shows the stack of the device. **b** A representative I–V curve with a current sweep mode showing two negative differential resistance (NDR) behaviors, noted to NDR-1 and NDR-2. The inset shows I–V curves in a voltage sweep mode with compliance currents ranging from 0.1 mA to 1 mA. **c** An oscillator circuit composed of the device and a series resistor. The TS device composed of a volatile memristor and an internal capacitor. **d** Experimentally obtained 20 oscillation current outputs (red and gray) under a fixed input voltage pulse (black). The left and right insets magnify the initial and final oscillation outputs, respectively. **e** A distribution of peak intervals ($t_{intvl}$) at each oscillation output. It shows 100 distributions collected from 100 testing cycles. Inset shows the average and standard deviation of the $t_{intvl}$ for 100 testing cycles.

follows[16].

$$\frac{dT}{dt} = \frac{i_m v_m}{C_{th}} - \frac{T - T_{amb}}{C_{th}R_{th}} + T\left(\frac{k_b}{C_{th}}\right)^{\frac{1}{2}} \frac{4\pi}{R_{th}C_{th}} \cos\left(\frac{2\pi t}{R_{th}C_{th}}\right), \quad (2)$$

where $T_{amb}$ is an ambient temperature, $C_{th}$ and $R_{th}$ are thermal capacitance and resistance respectively, and $t$ is a variable time constant describing the time variation in heat generation and dissipation. Here, we set the $t$ as a random variable to mimic the random thermal fluctuation.

Figure 2a shows the experimental and simulated current–voltage curves. Although there is some mismatch in the NDR-1 region, the model reproduced the experimental data quite well.

Then, the oscillation behavior of the oscillator circuit in Fig. 1c was numerically calculated under the same conditions as the experiments. Figure 2b shows 20 simulated oscillation waves obtained by the calculation, reproducing the experimental oscillation data in Fig. 1d very well. We assumed the device capacitance is 85 pF, and the parasitic capacitances are 42 pF. More detailed information regarding the circuit simulation can be found in Supplementary Fig. S1. Figure 2c shows the distribution of peak intervals in the simulated oscillation, which also reproduced the experimental results in Fig. 1e. The accurate simulation results confirm that the origin of the randomness is the thermal fluctuation.

Next, we conducted a finite element method (FEM)-based heat simulation to further understand the influence of thermal fluctuations on the stochastic oscillation, using the Field Triggered Thermal Runaway (FTTR) model in COMSOL[30]. Figure 2d shows the device geometry used in the simulation, referring to the device stack in Fig. 1a after electroforming. By the electroforming process, an oxygen-deficient filamentary conducting channel is formed[31–34]. Among the filament, a highly oxygen

deficient part (i.e., the anode interface) was crystallized to an insulating NbO2 as the NbO2 phase is thermodynamically stable. Accordingly, we assumed a filamentary Nb2O5−x region connected to an insulating NbO2 in series[30]. Then, the NbO2 is responsible for the TS characteristics by thermally activated transport mechanisms[20,21,35–37]. To mimic the oscillation, a sinusoidal voltage wave (0.4 MHz, 0.67–1.07 V) was applied to the top Pt electrode. Figure 2e shows the 20 consecutive switching curves under the sinusoidal voltage wave application. From point ① to ②, heat was generated in the NbO2. Then, the heat ran away along the Nb2O5−x filament while accelerating the thermally activated conduction in the NbO2 until point ③. Afterward, from point ③ to ④, the heat dissipated, and the device returned to point ①. Figure 2f shows the temperatures at point ② (red) and point ④ (blue) for each simulation cycle. The variation in switching temperature confirms that the residual heat was not a fixed value during the device oscillation. More detailed thermodynamic simulation results can be found in Supplementary Fig. S2.

**A self-clocking TRNG device.** The origin of the stochastic oscillation is the random thermal fluctuation. To extract the randomness from it, we developed a novel TRNG circuit embodying a self-clocking capability. The self-clocking characteristic is the prominent feature of the proposed TRNG. It can generate the essential clock signal by itself, enabling the TRNG to be compact and energy-efficient.

Figure 3a shows the TRNG circuit, including an input voltage source ($V_{pulse}$), a TS oscillator part (red square), a non-inverting amplifier part (blue square), and a negative edge triggered T flip-flop (green square). The 50 Ω of resistor represents the oscilloscope's internal resistance. In this configuration, the TS oscillator converts the constant voltage input at node 1 to an oscillating voltage output at node 2. The oscillating voltage output

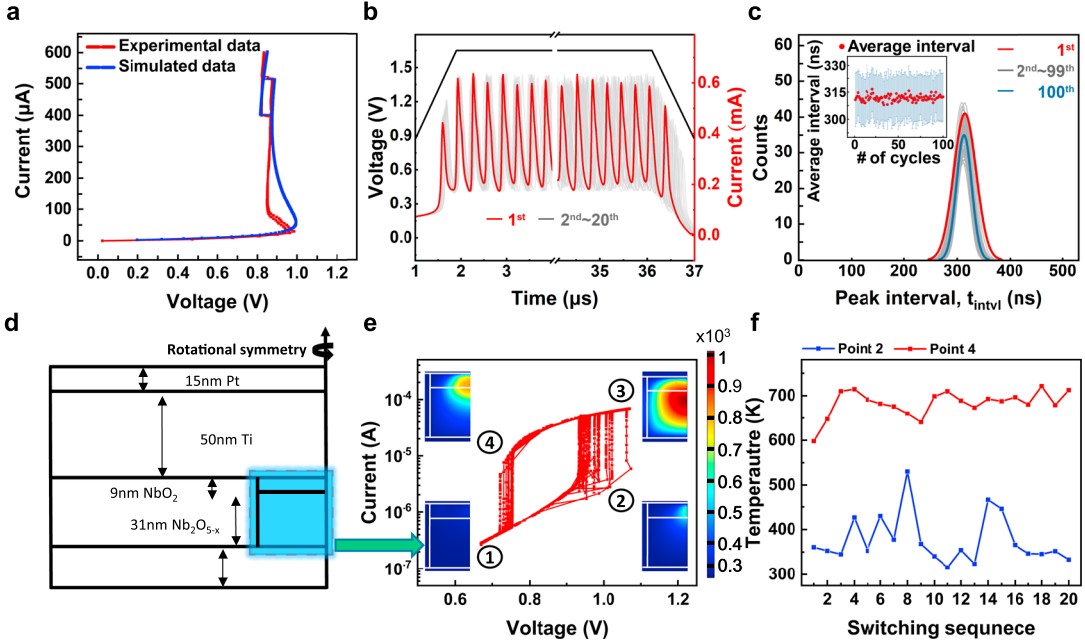

**Fig. 2 Simulations of stochastic oscillation characteristic induced by thermal fluctuation. a** A numerically simulated I–V curve (blue) compared with an experimental one (red line). **b** Simulated 20 oscillation currents (red and gray) under a fixed voltage input (black). The oscillation outputs overlap well at the beginning, but they disperse at the end. **c** Simulated distributions of the peak intervals for 100 cycles. Inset shows an average and a standard deviation of intervals. The results reproduce the experimental results in Fig. 1e. **d** A geometry used for a thermodynamic simulation, which is identical to the fabricated device in Fig. 1a. **e** Simulated I–V curves from the thermodynamic simulation under a sinusoidal input voltage. The insets show heat distribution at each moment. **f** The maximum temperatures at the moment of switch on (②, blue) and switch off (④, red) for 20 cycles of simulation, showing they fluctuate.

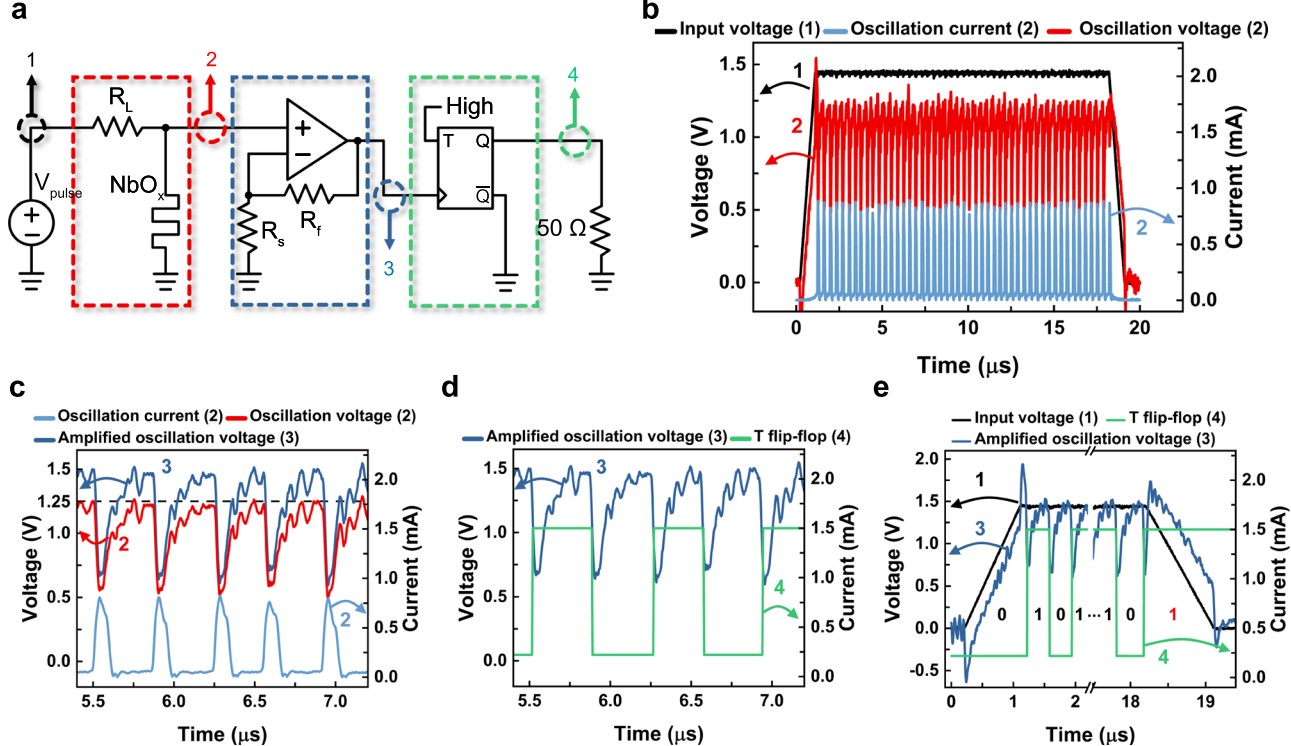

**Fig. 3 A TRNG circuit and its simulation. a** The proposed TRNG circuit composed of an input voltage source ($V_{pulse}$), a TS oscillator part (red square), a non-inverting amplifier part (blue square), and a negative edge triggered T flip-flop (green square). **b** The input voltage pulse (black) at node 1 and the output voltage (red) and current (bright blue) at node 2. **c** The amplified oscillation voltage at node 3 (blue) compared with the output voltage (red) and current (bright blue) at node 2. The horizontal line at 1.25 V indicates the required clock's peak voltage for T flip-flop operation. **d** The output current at node 4 (green) with respect to the amplified oscillation voltage as a clock of the T flip-flop. **e** The input voltage at node 1, the amplified voltage at node 3, and the output current at node 4 during the entire time frame for generating one random bit. In this case, the random output is 1.

ranges approximately from a low value of $V_h$ to a high value of $V_{th}$, where the $V_{th}$ and $V_h$ are the threshold and the hold voltages of the TS device, respectively. Also, the transition from high to low voltage is more drastic than the opposite transition because the heating rate is faster than the cooling rate[18]. Thus, we chose a negative edge triggered T flip-flop to utilize the drastic voltage drop as a clock signal. According to the T flip-flop's specification, its peak voltage should be higher than 1.25 V to be a valid clock signal. However, the $V_{th}$ was lower than 1.25 V with some variation, meaning it could not feed a stable clock signal. The non-inverting amplifier can pull up the input voltage by $(1 + R_f R_s^{-1})$ times to 1.25 V of the required clock voltage, where the $R_f$ and $R_s$ values are adjustable considering the $V_{th}$. Then, the T flip-flop utilizes the amplified oscillation voltage at node 3 for the clock signal.

Figure 3b shows the input voltage pulse at node 1 and the stochastic oscillation current and voltage at node 2 obtained by experiments. Figure 3c–e show the monitoring signals at nodes 2–4 in Fig. 3a obtained by a PSPICE simulation for validating the circuit operation. Figure 3c displays the oscillation current and voltage at node 2 and the amplified voltage at node 3. The horizontal line at 1.25 V indicates the required clock's peak voltage for T flip-flop operation. The op-amp amplifies the input voltage to trigger the subsequent flip-flop properly. If the oscillation voltage can be increased, the op-amp can be removed, and the device architecture can be more compact. Figure 3d shows the output current at node 4 with the clocking voltage at node 3. At every pulse falling moment, the flip-flop toggles its bit data. Figure 3e shows the entire signals of the input voltage, the amplified voltage, and the output current for generating one random bit. In the given period, the number of oscillation pulses is stochastically variable, but their probability of odd or even can be random, which allows true random number generation.

The self-clocking ability can be advantageous in energy as well. The energy consumption of the TS oscillator in Fig. 1c was 5.23 nJ bit$^{-1}$ ($V_A = 1.45$ V, $I_{avg} = 144$ μA, $t_{TRNG} = 25$ μs bit$^{-1}$), where $V_A$ is the applied operating voltage, $I_{avg}$ is the average current during operating, and $t_{TRNG}$ is the required time for true number generation. $I_{avg}$ is obtained by integrating the current during oscillation and dividing it by $t_{TRNG}$. It is higher than energy consumption of another TS memristors in previous TRNGs of 0.67 pJ bit$^{-1}$ ($V_A = 0.4$ V, $I_{avg} = 10^{-8}$ A, $t_{TRNG} = 166.6$ μs bit$^{-1}$)[6] and 0.63 pJ bit$^{-1}$ ($V_A = 10$ V, $I_{avg} = 10^{-9}$ A, $t_{TRNG} = 62.5$ μs bit$^{-1}$)[8]. However, they required an external clock generator that consumes about several hundred mW, which is much higher than the TS oscillator. (The power consumption of CDCI6214 by Texas Instruments, ultra-low-power clock generator, is 150 mW). Moreover, in our TRNG, less active components are used so that the total energy consumption can also be more efficient. More detailed energy comparison can be found in Supplementary Fig. S3.

**Experimental demonstration of the TRNG**. Figure 4a schematically shows the TRNG testing system configuration. The Keithley 4200A-SCS applied an input voltage pulse to the breadboard and received the output signal from it. Figure 4b shows the TRNG integrated on the breadboard where the op-amp, T flip-flop, and the resistors are shown (left panel). The right panel shows the corresponding circuit diagram. The non-inverting amplifier consisted of 1 kΩ $R_s$, 200 Ω $R_f$, and the op-amp (Model: NE5534D of Texas Instruments). The V+ terminal of the op-amp was connected to a 9 V battery and the V-terminal was grounded. The DC power supply was used to apply 2.5 V to the inputs J, K, (CLR), and $V_{CC}$ terminals of the flip-flop (Model: SN74LS73AN of Texas Instruments). The NbO$_x$ TS device was loaded in the probe station and connected to the breadboard via cables. In this setup, the parasitic capacitance in the breadboard is ~1 pF, and the op-amp and the flip-flop operate at a high speed of tens of ns. Thus, the signal delay by other components but the TS unit is negligible. More detailed discussions can be found in Supplementary Fig. S1.

Figure 4c shows the monitored input and output signals for five random bit generation cycling. The black, red, and green lines represent the input voltage pulse, the oscillation current of the device, and the output current, respectively. The input pulse amplitude and width were 1.1 V and 20 μs, respectively, with 1 μs of pulse rising and falling time. Although the oscillation peak current level was relatively low to 0.2 mA, it could provide sufficient voltage drop to the amplifier. The video recorded TRNG operation can be found in the Supplementary Video.

Figure 4d plots the $P$ value of the monobit test as a function of the input pulse period. Here, 100 random bits were used for the evaluation. The $P$ value can be obtained by following equation.

$$P\text{-value} = \text{erfc}\left(\sum_{i=1}^{n} \frac{|2\varepsilon_i - 1|}{\sqrt{2n}}\right), \qquad (4)$$

where $n$ and $\varepsilon_i$ are the length of the bit string and the datum of $i$th bit in the bit string respectively. The monobit test is the first test among the 15 tests in NIST (National Institute of Standards and Technology) random number test suite (NIST 800-22)[38] checking whether the fraction of ones in the given random bits is close to 1/2 or not, and it is the essential test because other tests would fail if the monobit test fails[38]. The recommended minimum data size for monobit test is 100 bits according to the instruction. The $P$ value is an indicator showing the randomness of the dataset; when the value is higher, the dataset is more random. In general, if the $P$ value exceeds 0.01, the dataset can be regarded as true random numbers.

When the time was short, the accumulation of peak interval time variations induced by thermal fluctuations was not sufficient, resulting in a low $P$ value. When the time was longer than ~10 μs, the $P$ value exceeded 0.01 continuously, guaranteeing that the output is a random number. This gives a random bit generation rate of 100 kb s$^{-1}$ for the monobit test. The random bit generation rate could be different by test types. For passing all 15 NIST randomness tests, we collected 130 sets of one Mbit (total 130 Mb) from the device using a 25 μs pulse (equivalent to 40 kb s$^{-1}$). Table 1 shows the testing results; the dataset passed all 15 NIST randomness tests successfully.

The endurance of our TRNG was about $4 \times 10^8$ cycles. Thus, it can generate at least $2.4 \times 10^7$ random bits per device. It is reasonably high for some practical applications such as security code generation in cryptography that require random number generation by request, not continuously[39]. More detailed endurance data can be found in Supplementary Fig. S4.

**Variation tolerance of the self-clocking TRNG device**. The thermal fluctuation results in a small peak interval time variation ($\triangle t_{intvl}$) between the oscillations. The $\triangle t_{intvl}$ is accumulated (i.e., added up) during oscillation. The proposed TRNG device utilizes the accumulation of thermal fluctuations as the randomness source. Actually, the amount of thermal fluctuations is time-variable and predictable, so that it cannot be random. However, by binarizing the thermal fluctuation and taking the remainder, the time-variant dimension can be eliminated, and a random output can be achieved (see Supplementary Fig. S5 for more discussion). This suggests that the device can be tolerant against any physical variation influencing the time-variable. Based on this observation, we further investigated the influence of two representative variations on device performance: a temperature variation and a device-to-device inherent performance variation.

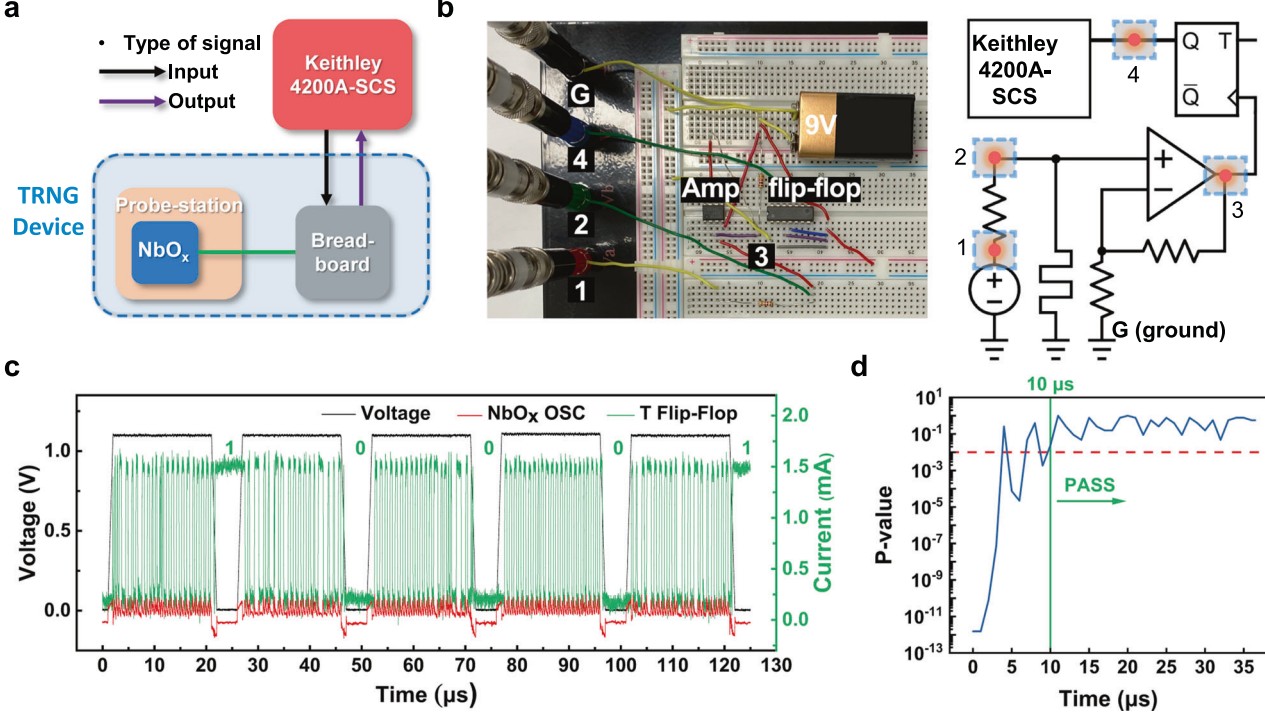

**Fig. 4 Experimental demonstration of the TRNG. a** An operating scheme of the proposed TRNG in experiments. **b** A photograph of the circuit built on a breadboard (left) and a circuit layout (right). **c** A demonstration of the random bit generation for six consecutive cycles. A black, red, and green lines show the input pulse voltage, the oscillation current, and the output current of the T flip-flop, respectively. **d** A $P$ value of the monobit test as a function of the input voltage pulse time. Above the $P$ value of 0.01 (red dashed line), it is regarded as random numbers. After 10 μs of pulse time, the outputs become true random numbers.

| Statistical test | $P$ value | Proportion | Result |
|---|---|---|---|
| 1. Frequency (monobit) test | 0.010751 | 1.0000 | PASS |
| 2. Frequency test within a block | 0.451595 | 0.9846 | PASS |
| 3. Runs test | 0.082824 | 0.9923 | PASS |
| 4. Test for the longest run of ones in a block | 0.066882 | 1.0000 | PASS |
| 5. Binary matrix rank test | 0.046361 | 1.0000 | PASS |
| 6. Discrete Fourier transform (spectral test) | 0.001173 | 0.9846 | PASS |
| 7. Non-overlapping template matching test | 0.095249 | 0.9846 | PASS |
| 8. Overlapping template matching test | 0.125088 | 1.0000 | PASS |
| 9. Maurer's "universal statistical" test | 0.017912 | 1.0000 | PASS |
| 10. Linear complexity test | 0.020984 | 1.0000 | PASS |
| 11. Serial test | 0.217681 | 1.0000 | PASS |
| 12. Approximate entropy test | 0.001106 | 0.9846 | PASS |
| 13. Cumulative sums (cusum) test | 0.003951 | 0.9769 | PASS |
| 14. Random excursions test | 0.009640 | 1.0000 | PASS |
| 15. Random excursions variant test | 0.001527 | 1.0000 | PASS |

**Table 1 The NIST test (800-22 test suite) results of the NbO$_x$-based TRNG.**

Figure 5a shows the current controlled NDR curves measured at different ambient temperatures ranging from 300 K to 390 K. Both the $V_{th}$ and $V_h$ decrease as the temperature increases, which is consistent with the previous report[40]. During the temperature testing, only the TS unit was heated in the probe station while other components were placed at room temperature. Above 390 K, the TS unit lost the NDR characteristic so that the

oscillation was not observed. Thus, 390 K was the maximum operating temperature of our device. Figure 5b shows the oscillation waves at each temperature. Under the same input voltage and load resistance conditions, the oscillation frequency increased due to the decrease in $V_{th}$ and $V_h$ with increasing ambient temperature.

Figure 5c shows the $P$ value of the NIST monobit test as a function of the pulse time at different temperatures. The inset summarizes the required time ($t_{TRNG}$) for the $P$ value to exceed 0.01, indicating a true random number, as a function of temperature. The TRNG worked well regardless of the ambient temperature. Moreover, as the ambient temperature increased, the random number generation speed became faster; at 300 K, the $P$ value exceeded 0.01 after 21.5 μs, giving a 46 kb s$^{-1}$ rate, while at 380 K, it was 15 μs, giving 66 kb s$^{-1}$. According to Eq. 2, the thermal fluctuation amplitude becomes higher with increasing ambient temperature. Therefore, the thermal fluctuation is more active at a higher temperature, resulting in a faster random number generation speed. The maximum bit generation rate can be as high as the oscillation frequency if the thermal fluctuation could be sufficiently high at the oscillating condition.

Figure 5d shows the current controlled NDR curves from five distinct devices, where they were intentionally manipulated for comparison. The load line by the $R_L$ of 3 k is included, suggesting all devices will oscillate at the given $R_L$. Figure 5e shows the oscillation waves from the devices under identical voltage pulse conditions. Despite the different oscillation frequencies, all of the devices generated true random numbers successfully.

Figure 5f shows the $P$ value of the NIST monobit test as a function of the pulse time for the different devices depicted in Fig. 5d. The inset plots the $t_{TRNG}$ of each device. Interestingly, the frequency of the oscillation was not directly correlated with the

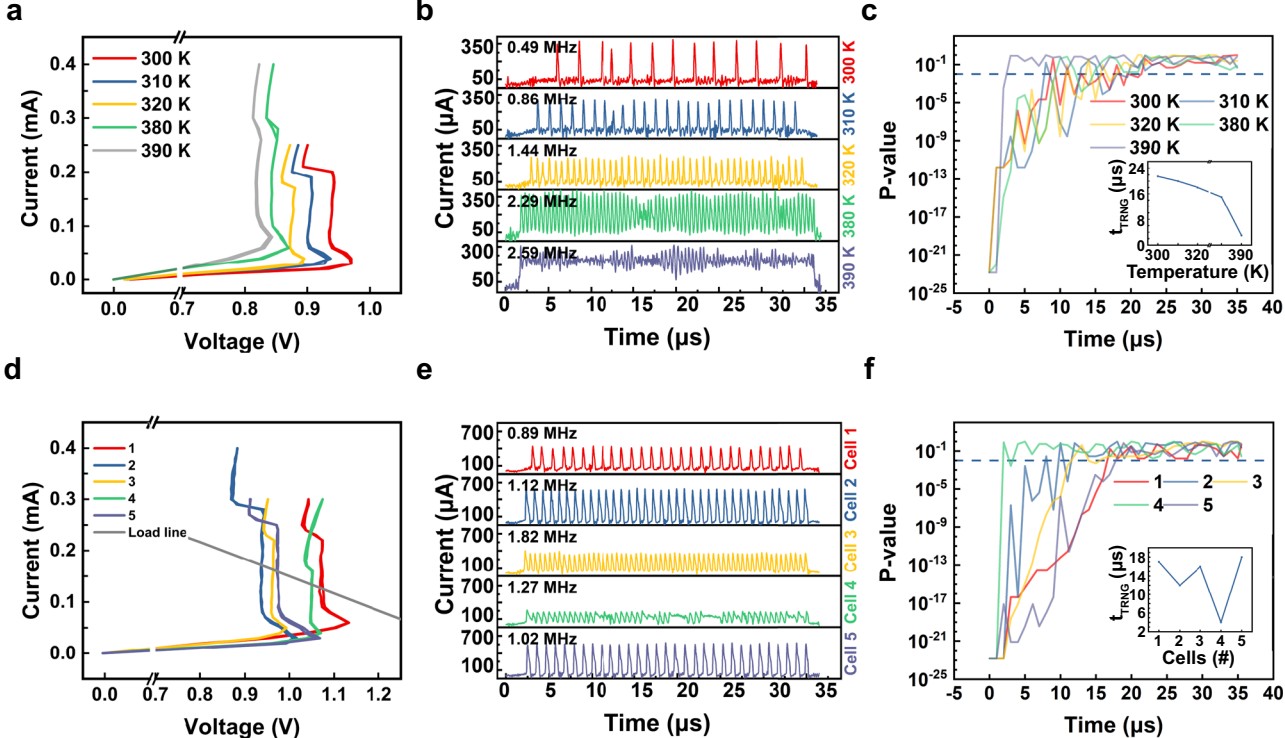

**Fig. 5 Variation tolerant test results at different ambient temperature and at different cells. a** The I–V curves at different temperatures from 300 K to 390 K from the same cell. **b** The oscillation current outputs at different temperatures. The frequency of oscillation increases as the temperature increases at the identical input voltage pulse. **c** The P value of the monobit test as a function of the input voltage pulse time using 100 bits per each temperature. The random bit reference line of 0.01 is included. The inset shows the minimum pulse time for random number generation as a function of the temperature. **d** Representatively distinguishable I–V curves from five different cells. **e** The oscillation current outputs and their frequency from each cell. **f** The P value for the monobit test obtained from 100 bits collected from each cell. They all pass the test at different random number generation time as shown in the inset.

$t_{TRNG}$, while the amplitude of the thermal fluctuation was closely correlated with the $t_{TRNG}$. The results suggest that the thermal fluctuation is proportional to the maximum temperature, but not to the temperature change. Therefore, as long as the TS devices oscillate, TRNG operation is possible, meaning the device is intrinsically variation tolerant.

In conclusion, we proposed a fast and highly tolerant TRNG device utilizing the stochastic self-oscillation behavior of the NbO$_x$ TS device. We found intrinsic stochasticity in the thermal fluctuation during the oscillation, and confirmed it by numerical and thermodynamic simulations. We developed a self-clocking TRNG circuit, extracting the random components by binarizing the thermal fluctuation, and successfully demonstrated the device operation by experiment. We collected 130 Mbits from the integrated device and confirmed that they passed all the NIST 800-22 random number tests. Also, we proved the variation tolerant characteristics of the device at various ambient temperatures and at various manipulated cells.

The proposed method utilizes a dimensionless quantity to generate the random number. Thus, as long as the device oscillates, the device can work, making the practical device application highly viable. Such high device reliability means it can be applied in various harsh environments in future electronics, wherever security is needed.

Lastly, the accumulation of the thermal fluctuation is proportional to time but independent of the frequency. Thus, even at the integrated device having fewer parasitic components and oscillating at a higher frequency, the bit generation rate will not be changed as long as the operating temperature is the same. For developing the higher speed of TRNG, embodying a heater near the TS unit can be plausible[41]. Also, some studies reported a chaotic oscillation behavior from the TS device at a controlled condition[16,17]. If such chaotic oscillation can be offered reliably, the bit generation rate could be as high as ~Mbit s$^{-1}$, equal to the oscillation frequency. This can be a promising perspective of research in the next.

## Methods

**Device fabrication**. A Pt/Ti/NbO$_x$/Pt device was fabricated using the following procedure. An adhesive 2-nm Ti followed by a 25-nm Pt bottom electrode was deposited by e-beam evaporation and patterned by a lift-off process on a SiO$_2$/Si substrate. Then, a 40-nm NbO$_x$ layer was deposited by a reactive sputtering process at 170 °C in Ar:O$_2$ (13:7, 4 mtorr) mixed gas ambient using Nb target. Then, a 50-nm Ti top electrode followed by a 15-nm Pt contact electrode was deposited by e-beam evaporation and patterned by a lift-off process.

**Electrical characterization**. In the TRNG circuit, a DC voltage was generated using EXO K-6135 power supply and 9 V battery. Also, the circuit components SN74LS73AN for JK flip-flop and NE55322P for op-amp made by Texas Instruments were used. All electrical characterization was performed using Keithley 4200A-SCS.

**Circuit simulation and NIST test**. The circuit simulation was executed by PSPICE. In PSPICE simulation, a model of u741 for amplifier and 7476 for JK flip-flop was used. The randomness test code was built in the Python environment, referring to the NIST Statistical Test Suite (Special Publication 800-22).

## Data availability

The data that support the findings of this study are available from the corresponding author upon reasonable request.

## Code availability

The code that supports the findings of this study is available from the corresponding author upon request.

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

## Acknowledgements

This research was supported by the MOTIE (Ministry of Trade, Industry & Energy) (Grant numbers 20003655 and 20003789) and KSRC (Korea Semiconductor Research Consortium) support program for the development of the future semiconductor device; and 2020 UP Research Project of KAIST.

## Author contributions

G.K. and J.H.I. contributed equally to this work. G.K. and J.H.I. conceived the idea, performed the experiments, and wrote the manuscript. Y.S.K., H.R., W.P., H.S., and J.P. helped with data analysis and discussed the results. K.M.K. supervised this study.

## Competing interests

The authors declare no competing interests.
