## [Peer Review File · Nature Communications]

Reviewers' Comments:

Reviewer #1:

Remarks to the Author:

The authors reported on a true random number generator (TRNG) based on the stochastic oscillation behavior of a NbOx Mott memristor, claiming the characteristics such as self-clocking, dimensionless values, fast generation rate up to 40 kb/s, and variation tolerant. They further attributed the source of randomness to the thermal fluctuation during oscillation. This work is novel, with most of the claims substantiated with experimental/simulation results. On the other hand, some issues should be addressed before being published.

1. What is the energy consumption landscape as compared with other types of TRNG? Both the current and voltage is fairly high for this device, as it relies on the thermal effect. If the NDR-2 type of effect is utilized, the device needs to be heated to a very high temperature (the metal-insulator-transition temperature for NbOx is over 1000K).
2. For a device to be used as TRNG, endurance is very important. However, there is no endurance data for the NbOx used in this work.
3. In Fig. 2, the device geometry used for simulation is different from the experimental description. The NbOx used in the simulation is a bi-layer structure with different compositions in those layers. Any explanation?
4. The authors showed that the TRNG is temperature tolerant with measurements up to 390 K, Why was this temperature range chose? Was it to accommodate some circuit operational temperature? It would be interesting to see the TRNG capability of the same device for a long time and monitor the junction temperature at the same time.
5. The oscillation frequency is determined by the RC. What determines the bit generation rate? It appears that a long enough input pulse is necessary. So, what is the fundamental limit of this generation rate and eventually the self-clocking capability?
6. The authors used an amplifier to enhance the amplitude of the signal before feeding it to the flip-flop (Fig. 3). Why is this amplifier necessary? If the voltage amplitude of the input signal is increased, would this amplifier still be necessary?

Reviewer #2:

Remarks to the Author:

This manuscript by Kim et al. reports a TRNG circuit based on the intrinsic thermal fluctuation in NbOx memristor. The TRNG circuit exhibits desirable features including self-clocking, fast bit generation and tolerance of temperature and device-to-device variations. Overall, there have been many existing approaches to building TRNG circuit based on memristors, and the uniqueness or significance of the present work is not very clear. In addition, this reviewer has the following questions:

1. Since the generation of the bit stream from the circuit depends on the oscillation, the endurance of the memristor will be an essential limit. Can authors show the endurance test data? Will the endurance be good enough to support such applications?
2. This work assumed a bilayer model consisting of 9 nm NbO₂ and 31 nm Nb₂O_{5-x} (Figure 2). Which is the switching layer that dominates the resistance state of the device? This is unclear to me. What is the meaning of a Nb₂O_{5-x} filament, given the physically more insulating of Nb₂O_{5-x}?
3. The TRNG circuit here utilizes the accumulation of thermal fluctuations, but the physical foundation of the accumulation is not studied or discussed. The authors should present careful study on the accumulation, as this forms the backbone for the circuit.
4. I found the references of the manuscript is not very complete. Related works on TRNG circuit,

especially those based on memristive devices, should be included.

Reviewer #3:

Remarks to the Author:

This manuscript reports a true random number generator (TRNG) utilizing stochastic oscillation characteristics of the NbOx Mott memristor. The authors identified the random source to be the thermal fluctuation. Based on their modeling and simulation, they built a breadboard-based system for true-random number generators and demonstrated a fast random bit rate. This could be a useful study, but several comments should be addressed before a recommendation is made.

First, there were several reports on utilizing the stochastic (or chaotic) nature of the NbOx device in various computing, including the cited ref. 12 and a recent one (doi: 10.1038/s41586-020-2735-5). The authors should discuss and compare the similarities and differences with the prior works.

Second, could the authors comment on the endurance performance of these threshold-switching devices? The high random number bit rate is highly dependent on the state switching of the device, which may impose a high requirement on the endurance.

The device capacitance plays a quite important role in the oscillation because it is induced by the RC delay. Could you author provide their estimation on the capacitance value? How about the parasitic capacitance in the breadboard?

If an on-chip integrated solution is provided in the future, will the result of the paper (such as bit rate due to different parasitic, power consumption, area overhead) change?

The NIST testing was collected with 25 μ s pulse (or 40kbs⁻¹), but in a different paragraph, the authors claim "This gives a random bit generation rate of 100 kbs⁻¹, which is the fastest record, compared with previous TS device-based TRNGs^{4,6,8}". What did not use the 100 kbs⁻¹ one for the NIST test?

Could the authors also comment on the bit rate dependence on the temperature? Fig. 5 shows the oscillation at 390 K is quite different from that at 300 K (room temperature). Does it mean it is close to the upper temperature limit of the device? How about low temperature?

Reviewer #1.

The authors reported on a true random number generator (TRNG) based on the stochastic oscillation behavior of a NbOx Mott memristor, claiming the characteristics such as self-clocking, dimensionless values, fast generation rate up to 40 kb/s, and variation tolerant. They further attributed the source of randomness to the thermal fluctuation during oscillation. This work is novel, with most of the claims substantiated with experimental/simulation results. On the other hand, some issues should be addressed before being published.

(Author's response)

Thank you for the careful reading of our manuscript. We prepared one-by-one responses to the reviewer's comments carefully.

#1. What is the energy consumption landscape as compared with other types of TRNG? Both the current and voltage is fairly high for this device, as it relies on the thermal effect. If the NDR-2 type of effect is utilized, the device needs to be heated to a very high temperature (the metal-insulator-transition temperature for NbOx is over 1000K).

(Author's response)

We agree that energy consumption is essential for the TRNG. Although our device utilizes the mott transition at a high temperature, the volume of NbO₂ filament is small so that the energy consumption for inducing the mott transition is not a problem.

To prove that, we evaluated the energy consumption of the previous volatile-based TRNG devices^{R1, R2}. In Ref. R1 by Jiang, H. *et al.*, (reference 6 in the main manuscript), the energy consumption can be obtained by the following equation; $E = V \times I \times (t_{\text{pulse}} - t_{\text{delay}})$, where V is the input voltage amplitude, I is the on-state current, t_{pulse} is the input pulse width, and t_{delay} is the average switching delay time. The calculated energy consumption is 0.8 pJ/bit (energy consumption at off-state is very small, 4×10^{-17} J/bit, which is negligible). Similarly, in ref. S2 by Woo, K. S. *et al.*, (reference 8 in the main manuscript) the energy consumption was about 3.15 pJ/bit based on the following properties; input voltage amplitude (10 V), on-state current (10^{-9} A), input voltage width (350 μ s), average switching delay time at 10 V (35 μ s).

In our device, the energy consumption can be calculated by integrating the oscillation current during the random number generating time (25 μ s, please refer to Fig. 5e) and multiplying the input voltage (1.45 V). It gives an energy consumption of about 5.23 nJ/bit.

It seems that our TRNG unit consumes more energy than other TRNG devices.

However, considering external circuit elements including the clock generator, the energy consumption of our device can be a way lower. In ref. R1, it required a clocking unit and a typical ultra-low-power clock generator's (e.g., CDCI6214, Texas Instruments) power consumption is 150 mW. Considering the bit generation time is 300 μ s, the clock generator solely consumes 45 μ J/bit. It reveals that the self-clocking ability is advantageous not only for circuit simplification but also for energy efficiency.

We added a discussion regarding the energy consumption in page 8 as follows.

“The self-clocking ability can be advantageous in energy as well. The energy consumption of the TS memristor itself was 5.23 nJ/bit ($V_A = 1.45$ V, $I_{avg} = 144$ μ A, $t_{TRNG} = 25$ μ s), where V_A is the applied operating voltage, I_{Avg} is the average current during operating, and t_{TRNG} is the required time for true number generation. It is higher than an energy consumption of another TS memristors in previous TRNGs of 0.8 pJ/bit ($V_A = 0.5$ V, $I_{avg} = 10^{-8}$ A, $t = 200$ μ s)⁶ and 3.15 pJ/bit ($V = 10$ V, $I = 10^{-9}$ A, $t = 350$ μ s)⁸. However, they required active external components such as clock generators that consume about several hundred mW, much higher than the TS unit. (The power consumption of CDCI6214 by Texas Instruments, ultra-low-power clock generator, is 150 mW). In our TRNG, no active components are needed so that the total energy consumption can also be efficient.”

- R1 Jiang, H. *et al.* A novel true random number generator based on a stochastic diffusive memristor. *Nat Commun* **8**, 882 (2017).
- R2 Woo, K. S. *et al.* A Combination of a Volatile Memristor-Based True Random Number Generator and a Nonlinear Feedback Shift Register for High-Speed Encryption. *Advanced Electronic Materials* **6**, 1901117 (2020).

#2. For a device to be used as TRNG, endurance is very important. However, there is no endurance data for the NbO_x used in this work.

(Author's response)

We also agree that endurance is an essential characteristic. The endurance of the NbO_x TS memristor is typically very high as $\sim 6.5 \times 10^{10}$ in Ref. R3, and $\sim 10^{12}$ in Ref. R4. Our NbO_x-based mott memristor could also switch more than 4×10^8 cycles and so it could generate 2.4×10^7 random bits. The endurance could be far more improved after device optimization.

In the revised manuscript, we included the endurance data in supplementary information Figure S3. We revised the manuscript and the figure caption accordingly.

In page 10,

“The endurance of our TRNG was about 4×10^8 cycles. Thus, it can generate at least 2.4×10^7 random bits per device, which is reasonably high for the practical application. More detailed endurance data can be found in **Supplementary Fig. S3.**”

In Supplementary Fig. S3.

Supplementary Figure S3 | Endurance characteristics of the NbO_x-based mott device. For endurance testing, we applied a constant DC voltage to the NbO_x-based oscillator and monitored the oscillating pulses per every 40 seconds with a load resistance of 4 kΩ. **a** shows three pulse trains (reading – cycling – reading) whose widths are 20 μs, 40 s, and 20 μs, respectively. The pulse amplitude was 2.2 V. The reading pulse verifies whether the oscillator works or not. The cycling pulse operates the device for several million times approximately. The oscillation data at the reading period after **b** 40 s, **c** 120 s, **d** 240 s, **e** 480 s, and **f** 600 s are shown. The number of switching was calculated by the following equation: $f_{avg,s} \times t$, where $f_{avg,s}$ is the average oscillation frequency of 0.67 MHz, and t is 600 seconds. It gives 4×10^8 cycles. Our TRNG can generate one bit per 25 μs. Thus, it can generate at least 2.4×10^7 random bits per device, which is reasonably high for the practical application.

- R3 Li, S. *et al.* High-endurance megahertz electrical self-oscillation in Ti/NbO_x bilayer structures. *Applied Physics Letters* **106**, 212902 (2015)
- R4 Zhang, X. *et al.* An artificial spiking afferent nerve based on Mott memristors for neurorobotics. *Nat Commun* **11**, 51 (2020).

#3. In Fig. 2, the device geometry used for simulation is different from the experimental description. The NbO_x used in the simulation is a bi-layer structure with different

compositions in those layers. Any explanation?

(Author's response)

At first, the NbO_x layer is homogeneous in an amorphous phase. The device is then electroformed by applying -5 V to the top electrode with a 5mA compliance current. During this electroforming step, a filamentary conducting channel is formed in the device^{R5-R8}. The conducting channel consists of two parts serially; a conductive Nb₂O_{5-x} filament region and an insulating NbO₂ region. The NbO₂ formation can be understood as follows; a highly oxygen deficient part of the filaments (i.e., the anode interface) was crystallized to an insulating NbO₂ as the NbO₂ phase is thermodynamically stable. Then, the NbO₂ shows a volatile switching characteristic by thermally active transport like Poole-Frenkel emission^{R9-R13}. The device geometry used in Fig. 2 refers to the device geometry after electroformed where the NbO₂ and Nb₂O_{5-x} coexists^{R7, R14, R15}.

We revised the paragraph regarding Fig. 2d in page 6 for better description as follows.

“Figure 2d shows the device geometry used in the simulation, referring to the device stack in Fig. 1a after electroforming. By the electroforming process, an oxygen-deficient filamentary conducting channel is formed³¹⁻³⁴. Among the filament, a highly oxygen deficient part (i.e., the anode interface) was crystallized to an insulating NbO₂ as the NbO₂ phase is thermodynamically stable. Accordingly, we assumed a filamentary Nb₂O_{5-x} region connected to an insulating NbO₂ in series³⁵. Then, the NbO₂ is responsible for the TS characteristics by thermally-activated transport mechanisms³⁶⁻⁴⁰. To mimic the oscillation, a sinusoidal voltage wave (0.4 Mhz, 0.67 V – 1.07 V) was applied to the top Pt electrode.”

- R5 Slesazeck, S. et al. Physical model of threshold switching in NbO₂ based memristors. *RSC Adv* **5**, 102318–102322 (2015).
- R6 Chudnovskii, F. A. et al. Electroforming and switching in oxides of transition metals: The role of metal-insulator transition in the switching mechanism. *Journal of Solid State Chemistry* **122**, 95–99 (1996).
- R7 Nandi, S. K. et al. Threshold current reduction for the metal–insulator transition in NbO_{2-x}-selector devices: the effect of ReRAM integration. *Applied Physics Letters* **48**, 195105 (2015).
- R8 Luo, Q. et al. Memory switching and threshold switching in a 3D nanoscaled NbO_x system. *IEEE Electron Device Lett* **40**, 718–721 (2019).
- R9 Shin, S. H., Halpern, T. & Raccach, P. M. High-speed high-current field switching of NbO₂. *J. Appl. Phys* **48**, 3150–3153 (1977).
- R10 Kumar, S. et al. Physical origins of current and temperature controlled negative differential resistances in NbO₂. *Nat. Commun* **8**, 658 (2017).

- R11 Gibson, G. A. et al. An accurate locally active memristor model for S-type negative differential resistance in NbOx. *Applied Physics Letters* **108**, 023505 (2016).12
- R12 Goodwill, J. M. et al. Spontaneous current constriction in threshold switching devices. *Nat. Commun* **10**, 1628 (2019).
- R13 Cha, E. et al. Nanoscale (~10 nm) 3D vertical ReRAM and NbO2 threshold selector with TiN electrode. In 2013 *IEEE International Electron Devices Meeting (IEDM)* **14062129**, 10.5.1–10.5.4 (IEEE, 2013).
- R14 Funck, C. et al. Multidimensional simulation of threshold switching in NbO2 based on an electric field triggered thermal runaway model. *Adv. Electron. Mater* **2**, 1–13 (2016).
- R15 Li, S., Liu, X., Nandi, S. K. & Elliman, R. G. Anatomy of filamentary threshold switching in amorphous niobium oxide. *Nanotechnology* **29**, 375705 (2018).

#4. The authors showed that the TRNG is temperature tolerant with measurements up to 390 K, Why was this temperature range chose? Was it to accommodate some circuit operational temperature? It would be interesting to see the TRNG capability of the same device for a long time and monitor the junction temperature at the same time.

(Author's response)

With increasing the ambient temperature, both the threshold voltage (V_{th}) and hold voltage (V_h) decrease, and the V_{th} decrease is more significant. Eventually, the mott memristor loses its threshold switching characteristics as the NDR disappears at the elevated temperature above 400 K^{R16}. Thus, 390 K was the maximum operating temperature of the TS device.

We elevated only the TS device temperature by heating the holder in the probe station, leaving the breadboard at room temperature. In general, the operating temperature range of the commercial electrical components is lower than 390 K; those of the op-amp and the JK flip-flop are below 343 K. Thus, it seems not likely possible to put all components at such high temperature. Therefore, if one wants to operate the TS device at a high temperature in the integrated device, a careful chip design will be necessary to relieve the heat issues.

We elaborated and clarified the temperature issues in page 10 as below.

“Figure 5a shows the current controlled NDR curves measured at different ambient temperatures ranging from 300 K to 390 K. Both the V_{th} and V_h decrease as the temperature increases, which is consistent with the precious reprot⁴². During the temperature testing, only the TS unit was heated in the probe station while other components were placed at room temperature. Above 390 K, the TS unit lost the NDR characteristic so that the oscillation was not observed. Thus, 390 K was the maximum operating temperature of our device.

Figure 5b shows the oscillation waves at each temperature.”

R16 Nandi, S. K., Li, S., Liu, X. & Elliman, R. G. Temperature dependent frequency tuning of NbOx relaxation oscillators. *Applied Physics Letters* **111**, 202901 (2017).

#5. The oscillation frequency is determined by the RC. What determines the bit generation rate? It appears that a long enough input pulse is necessary. So, what is the fundamental limit of this generation rate and eventually the self-clocking capability?

(Author's response)

As the reviewer correctly mentioned, the oscillation frequency is mainly associated with the RC delay. What we argue in this study is that the RC delay is fluctuating during every oscillation due to the internal temperature fluctuation. It results in a peak interval time variation stochastically.

The stochastic time interval variation is added up over oscillation cycles. Let Δt_{intvl} is the standard deviation of the peak interval time, and t_{intvl} is an average peak interval time. Then, the peak interval between two peaks can be expressed as $t_{intvl} + \Delta t_{intvl}$. We can estimate the n th oscillation peak will appear at $n \times (t_{intvl} + \Delta t_{intvl})$. It means that in the n th oscillation, the accumulated time interval variation can be $n \times \Delta t_{intvl}$. If the accumulated time interval variation is longer than the time interval ($\frac{n \times \Delta t_{intvl}}{t_{intvl}} > 1$), at time of $n \times t_{intvl}$ there is a chance that the accumulated time interval variation can exceed the average time interval. It is the regime where our TRNG operates.

We clarified the discussion on the thermal fluctuation accumulation as follows.

In page 4.

“During the oscillation, the capacitive components in the device influence the frequency of the oscillation by inducing an RC delay, which is dependent on temperatures. Thus, the heat generation and dissipation conditions are stochastic, resulting in a slight variation in the oscillation period.”

In page 9,

“When the time was short, the accumulation of of peak interval time variations induced by thermal fluctuations was not sufficient, resulting in a low P-value. When the time was longer than $\sim 10 \mu\text{s}$, the P-value exceeded 0.01 continuously, guaranteeing that the output is a random number.”

In page 10,

“The thermal fluctuation results in a small peak interval time variation (Δt_{intvl}) between the oscillations. The Δt_{intvl} is accumulated (i.e., added up) during oscillation. The proposed TRNG device utilizes the accumulation of Δt_{intvl} by thermal fluctuations as the randomness source. Actually, the amount of thermal fluctuations is time-variable and predictable, so that it cannot be random.”

Regarding the fundamental limit of the generation rate, it depends on the Δt_{intvl} . If the Δt_{intvl} is longer than t_{intvl} , the random bit generation will be possible per every oscillation ideally. Thus, in this case, the fundamental limit can be equal to the oscillation frequency.

We added a brief discussion on the limitation of rate as below.

In page 11,

“The maximum bit generation rate can be as high as the oscillation frequency if the thermal fluctuation could be sufficiently high at the oscillating condition.”

#6. The authors used an amplifier to enhance the amplitude of the signal before feeding it to the flip-flop (Fig. 3). Why is this amplifier necessary? If the voltage amplitude of the input signal is increased, would this amplifier still be necessary?

(Author's response)

Yes, the review is correct. The oscillation amplitude was slightly weak to operate the T flip-flop. As the reviewer mentioned, if the voltage amplitude of the input signal is increased enough to trigger the T flip-flop, the amplifier may not be necessary. Then, the TRNG circuit can be more compact.

We elaborated it in page 7 as follows.

“Figure 3c displays the currents at node 2 and node 3. The oscillation current was weak to operate the T flip-flop directly. The op-amp amplifies the input signal to trigger the subsequent flip-flop properly. If the oscillation current can be increased, the op-amp can be removed, and the device architecture can be more compact.”

Reviewer #2.

This manuscript by Kim et al. reports a TRNG circuit based on the intrinsic thermal fluctuation in NbOx memristor. The TRNG circuit exhibits desirable features including self-clocking, fast bit generation and tolerance of temperature and device-to-device variations. Overall, there have been many existing approaches to building TRNG circuit based on memristors, and the uniqueness or significance of the present work is not very clear.

(Author's response)

We thank the reviewer for the valuable comments. Through this response letter, we hope we can correctly answer the reviewer's concerns and clarify our arguments better.

#1. Since the generation of the bit stream from the circuit depends on the oscillation, the endurance of the memristor will be an essential limit. Can authors show the endurance test data? Will the endurance be good enough to support such applications?

(Author's response)

In the revised manuscript, we included the endurance data in supplementary information Fig. S3. We revised the manuscript accordingly.

In page 10,

“The endurance of our TRNG was about 4×10^8 cycles. Thus, it can generate at least 2.4×10^7 random bits per device, which is reasonably high for the practical application. More detailed endurance data can be found in **Supplementary Fig. S3.**”

In Figure S3,

Supplementary Figure S3 | Endurance characteristics of the NbO_x-based mott device. For endurance testing, we applied a constant DC voltage to the NbO_x-based oscillator and monitored the oscillating pulses per

every 40 seconds with a load resistance of 4 kΩ. a shows three pulse trains (reading – cycling – reading) whose widths are 20 μs, 40 s, and 20 μs, respectively. The pulse amplitude was 2.2 V. The reading pulse verifies whether the oscillator works or not. The cycling pulse operates the device for several million times approximately. The oscillation data at the reading period after b 40 s, c 120 s, d 240 s, e 480 s, and f 600 s are shown. The number of switching was calculated by the following equation: $f_{avg,s} \times t$, where $f_{avg,s}$ is the average oscillation frequency of 0.67 MHz, and t is 600 seconds. It gives 4×10^8 cycles. Our TRNG can generate one bit per 25 μs. Thus, it can generate at least 2.4×10^7 random bits per device, which is reasonably high for the practical application.

The endurance of the NbO_x memristor is known to be greater ($\sim 6.5 \times 10^{10}$ in Ref R1, and $\sim 10^{12}$ in Ref R2). Our NbO_x-based mott memristor could switch more than 4×10^8 and could generate 24×10^6 random bits. The endurance could be improved with device optimization if necessary.

- R1 Li, S. *et al.* High-endurance megahertz electrical self-oscillation in Ti/NbO_x bilayer structures. *Applied Physics Letters* **106**, 212902 (2015)
- R2 Zhang, X. *et al.* An artificial spiking afferent nerve based on Mott memristors for neurorobotics. *Nat Commun* **11**, 51 (2020).

#2. This work assumed a bilayer model consisting of 9 nm NbO₂ and 31 nm Nb₂O_{5-x} (Figure 2). Which is the switching layer that dominates the resistance state of the device? This is unclear to me. What is the meaning of a Nb₂O_{5-x} filament, given the physically more insulating of Nb₂O_{5-x}?

(Author's response)

The initial matrix of our device was an insulating Nb₂O₅. During the electroforming process, a conducting path was formed, which is composed of an oxygen-deficient Nb₂O_{5-x}. Then, some of the portions of filament containing the higher oxygen defect concentration was crystallized to an insulating NbO₂ phase since it is thermodynamically more stable^{R3-R8}. It resulted in a serial configuration of Nb₂O_{5-x} and NbO₂ as described in Fig. 2. Then, the NbO₂ region is responsible for the threshold switching^{R9-R13}. Oxygen vacancies in Nb₂O_{5-x} are known to be n-type dopants^{R14-R17} so that the Nb₂O_{5-x} region acts as a resistor.^{R18, R19}

We elaborated it in page 6 as below.

“Figure 2d shows the device geometry used in the simulation, referring to the device stack in Fig. 1a after electroforming. By the electroforming process, an oxygen-deficient filamentary conducting channel is formed³¹⁻³⁴. Among the filament, a highly oxygen deficient part (i.e., the anode interface) was crystallized to an insulating NbO₂ as the NbO₂ phase is

thermodynamically stable. Accordingly, we assumed a filamentary Nb₂O_{5-x} region connected to an insulating NbO₂ in series³⁵. Then, the NbO₂ is responsible for the TS characteristics by thermally-activated transport mechanisms³⁶⁻⁴⁰. To mimic the oscillation, a sinusoidal voltage wave (0.4 Mhz, 0.67 V – 1.07 V) was applied to the top Pt electrode.”

- R3 Slesazek, S. et al. Physical model of threshold switching in NbO₂ based memristors. *RSC Adv* **5**, 102318–102322 (2015).
- R4 Chudnovskii, F. A. et al. Electroforming and switching in oxides of transition metals: The role of metal-insulator transition in the switching mechanism. *Journal of Solid State Chemistry* **122**, 95–99 (1996).
- R5 Nandi, S. K. et al. Threshold current reduction for the metal–insulator transition in NbO_{2-x} -selector devices: the effect of ReRAM integration. *Applied Physics Letters* **48**, 195105 (2015).
- R6 Funck, C. et al. Multidimensional simulation of threshold switching in NbO₂ based on an electric field triggered thermal runaway model. *Adv. Electron. Mater* **2**, 1–13 (2016).
- R7 Li, S., Liu, X., Nandi, S. K. & Elliman, R. G. Anatomy of filamentary threshold switching in amorphous niobium oxide. *Nanotechnology* **29**, 375705 (2018).
- R8 Luo, Q. et al. Memory switching and threshold switching in a 3D nanoscaled NbO_x system. *IEEE Electron Device Lett* **40**, 718–721 (2019).
- R9 Shin, S. H., Halpern, T. & Raccach, P. M. High-speed high-current field switching of NbO₂. *J. Appl. Phys* **48**, 3150–3153 (1977).
- R10 Kumar, S. et al. Physical origins of current and temperature controlled negative differential resistances in NbO₂. *Nat. Commun* **8**, 658 (2017).
- R11 Gibson, G. A. et al. An accurate locally active memristor model for S-type negative differential resistance in NbO_x. *Applied Physics Letters* **108**, 023505 (2016).¹²
- R12 Goodwill, J. M. et al. Spontaneous current constriction in threshold switching devices. *Nat. Commun* **10**, 1628 (2019).
- R13 Cha, E. et al. Nanoscale (~10 nm) 3D vertical ReRAM and NbO₂ threshold selector with TiN electrode. In 2013 *IEEE International Electron Devices Meeting (IEDM)* **14062129**, 10.5.1–10.5.4 (IEEE, 2013).
- R14 Nunes, B. N., Lopes, O. F., Patrocinio, A. O. T. & Bahnemann, D. W. Recent advances in niobium-based materials for photocatalytic solar fuel production. *Catalysts* **10**, (2020).
- R15 Kurmaev, E. Z. et al. Electronic structure of niobium oxides. *Journal of Alloys and Compounds*. **347** 213–218 (2002).

- R16 Sikula, J., Hlavka, J., Sedlakova, V. & Grmela, L. Conductivity Mechanisms and Breakdown Characteristics of Niobium Oxide Capacitors. *Carts* 0–4 (2003).
- R17 Ninomiya, T. et al. Conductive filament scaling of TaO x bipolar ReRAM for long retention with low current operation. in Digest of Technical Papers - *Symposium on VLSI Technology* 73–74 (2012).
- R18 Yang, J. J. et al. Memristive switching mechanism for metal/oxide/metal nanodevices. *Nature Nanotechnology* **3**, 429–433 (2008).
- R19 Nico, C., Monteiro, T. & Graça, M. P. F. Niobium oxides and niobates physical properties: Review and prospects. *Progress in Materials Science* **80**, 1–37 (2016).

#3. The TRNG circuit here utilizes the accumulation of thermal fluctuations, but the physical foundation of the accumulation is not studied or discussed. The authors should present careful study on the accumulation, as this forms the backbone for the circuit.

(Author's response)

Thank you for your careful reading of our manuscript. The proposed TRNG device utilized intrinsic thermal fluctuation during oscillation. This thermal fluctuation caused a variation in the period between the oscillations, i.e., a peak interval time variation (Δt_{intvl}). The Δt_{intvl} was accumulated (or added up) during oscillation, so we expressed it as an ‘accumulation of thermal fluctuations’. As shown in inset of Fig. 1d, the oscillation waves were almost overlapped at the beginning because the total sum of Δt_{intvl} was yet small. However, as the total sum of Δt_{intvl} get more significant over time, it can exceed the average time interval and the oscillation gets unpredictable.

We clarified the discussion on the thermal fluctuation accumulation as follows.

In page 4.

“During the oscillation, the capacitive components in the device influence the frequency of the oscillation by inducing an RC delay, which is dependent on temperatures. Thus, the heat generation and dissipation conditions are stochastic, resulting in a slight variation in the oscillation period.”

In page 9,

“When the time was short, the accumulation of peak interval time variations induced by thermal fluctuations was not sufficient, resulting in a low P-value. When the time was longer than $\sim 10 \mu\text{s}$, the P-value exceeded 0.01 continuously, guaranteeing that the output is a random number.”

In page 10,

“The thermal fluctuation results in a small peak interval time variation ($\Delta t_{interval}$) between the oscillations. The $\Delta t_{interval}$ is accumulated (i.e., added up) during oscillation. The proposed TRNG device utilizes the accumulation of $\Delta t_{interval}$ by thermal fluctuations as the randomness source. Actually, the amount of thermal fluctuations is time-variable and predictable, so that it cannot be random.”

#4. I found the references of the manuscript is not very complete. Related works on TRNG circuit, especially those based on memristive devices, should be included.

(Author's response)

Thanks for the comment. We investigated memristor-based TRNG studies thoroughly and revised our manuscript as below.

In page 2,

“For better security, researchers have proposed novel methods for realizing a TRNG, utilizing the stochastic switching devices such as spin transfer-torque, magnetic memory, and resistive memory²⁻¹⁵.”

- R9 Gaba, S., Sheridan, P., Zhou, J., Choi, S. & Lu, W. Stochastic memristive devices for computing and neuromorphic applications. *Nanoscale* **5**, 5872-5878 (2013).
- R10 Fukushima, A. et al. Spin dice: A scalable truly random number generator based on spintronics. *Applied Physics Express* **7**, 083001 (2014).
- R11 Won Ho, C. et al. in *IEEE International Electron Devices Meeting*. 12.15.11-12.15.14 (2014).
- R12 Wang, Y., Wen, W., Li, H. & Hu, M. in *Proceedings of the 25th edition on Great Lakes Symposium on VLSI* 271-276 (2015).
- R13 Hashim, N. A. B. N., Teo, J., Hamid, M. S. A. & Hamid, F. A. B. in *IEEE Student Conference on Research and Development (SCOReD)*. 1-5 (2016).
- R14 Zhang, T. et al. High-speed true random number generation based on paired memristors for security electronics. *Nanotechnology* **28**, 455202 (2017).
- R15 Rai, V. K., Tripathy, S. & Mathew, J. Memristor based Random Number Generator: Architectures and Evaluation. *Procedia Computer Science* **125**, 576-583 (2018).

Reviewer #3.

This manuscript reports a true random number generator (TRNG) utilizing stochastic oscillation characteristics of the NbOx Mott memristor. The authors identified the random source to be the thermal fluctuation. Based on their modeling and simulation, they built a breadboard-based system for true-random number generators and demonstrated a fast random

bit rate. This could be a useful study, but several comments should be addressed before a recommendation is made.

(Author's response)

We thank you for your valuable comments. We prepared one-by-one responses carefully.

#1. There were several reports on utilizing the stochastic (or chaotic) nature of the NbO_x device in various computing, including the cited ref. 12 and a recent one (doi: 10.1038/s41586-020-2735-5). The authors should discuss and compare the similarities and differences with the prior works.

(Author's response)

Thanks for the comment. We added one paragraph as below in the introduction, revoking the importance of NbO₂ devices.

In page 2,

"A NbO_x-based mott memristor shows a stochastic (or chaotic in specific condition) threshold switching behavior. Various applications have been proposed to exploit it such as the Hopfield network or fundamental oscillatory computing^{16,17}. Considering such high usefulness of the NbO₂ mott memristor for various applications, its application to the TRNG should be reasonable, but no studies have demonstrated it yet.

”

R1 Kumar, S., Strachan, J. P. & Williams, R. S. Chaotic dynamics in nanoscale NbO₂ Mott memristors for analogue computing. *Nature* **548**, 318-321 (2017).

R2 Kumar, S., Williams, R. S. & Wang, Z. Third-order nanocircuit elements for neuromorphic engineering. *Nature* **585**, 518–523 (2020).

#2. Could the authors comment on the endurance performance of these threshold-switching devices? The high random number bit rate is highly dependent on the state switching of the device, which may impose a high requirement on the endurance.

(Author's response)

Thank you for the comment. In the revised manuscript, we included the endurance data in supplementary information Fig. S3. We revised the manuscript accordingly.

In page 10,

“The endurance of our TRNG was about 4×10^8 cycles. Thus, it can generate at least 2.4×10^7 random bits per device, which is reasonably high for the practical application. More detailed endurance data can be found in **Supplementary Fig. S3.**”

In Figure S3,

Supplementary Figure S3 | Endurance characteristics of the NbO_x-based mott device. For endurance testing, we applied a constant DC voltage to the NbO_x-based oscillator and monitored the oscillating pulses per every 40 seconds with a load resistance of 4 kΩ. **a** shows three pulse trains (reading – cycling – reading) whose widths are 20 μs, 40 s, and 20 μs, respectively. The pulse amplitude was 2.2 V. The reading pulse verifies whether the oscillator works or not. The cycling pulse operates the device for several million times approximately. The oscillation data at the reading period after **b** 40 s, **c** 120 s, **d** 240 s, **e** 480 s, and **f** 600 s are shown. The number of switching was calculated by the following equation; $f_{avg,s} \times t$, where $f_{avg,s}$ is the average oscillation frequency of **0.67** MHz, and t is 600 seconds. It gives 4×10^8 cycles. Our TRNG can generate one bit per 25 μs. Thus, it can generate at least 2.4×10^7 random bits per device, which is reasonably high for the practical application.

The endurance of the NbO_x memristor is known to be greater ($\sim 6.5 \times 10^{10}$ in Ref R3, and $\sim 10^{12}$ in Ref R4). Our NbO_x-based mott memristor could switch more than 4×10^8 and could generate 24×10^6 random bits. The endurance could be improved with device optimization if necessary.

R3 Li, S. *et al.* High-endurance megahertz electrical self-oscillation in Ti/NbO_x bilayer structures. *Applied Physics Letters* **106**, 212902 (2015)

R4 Zhang, X. *et al.* An artificial spiking afferent nerve based on Mott memristors for neurorobotics. *Nat Commun* **11**, 51 (2020).

#3. The device capacitance plays a quite important role in the oscillation because it is induced by the RC delay. Could you author provide their estimation on the capacitance value? How about the parasitic capacitance in the breadboard?

(Author's response)

We agree that the RC delay plays an important role in the oscillation. We discussed more details in supplementary information Fig. S4 as below.

Supplementary Figure S4 | Oscillator equivalent circuit configuration. For oscillation simulation, we used this circuit configuration. Here, C_{p1} and C_{p2} are the parasitic capacitances by electrical measurement setup or device geometry. The C_{device} is the NbO_x TS device capacitance, R_{osc} is a 50 ohms-resistor in the oscilloscope, and the R_L is the load resistor connected with the device in series. For the best fitting, we used the capacitance values, 42pF, 42pF, and 85pF for C_{p1} , C_{p2} , and C_{device} , respectively.

The parasitic capacitance in the breadboard is ~ 1 pF which is very low compared to the capacitors above. The specification of op-amp says, it will take 40 ns for it to successfully amplify the input. (We used NE55322P op-amp having 9V/ μ s slew rate. Typical ΔV_{NDR} ($V_{th}-V_h$) is ~ 0.15 V, so the amplifier should change its output $2\Delta V_{NDR}$ multiplied by 1.2 which is the amplification factor determined by $1+R_f/R_s$ ($R_f = 200\Omega$, $R_s = 1k\Omega$ in Fig. 4). Also, it takes about 16 ns for a typical flip-flop to convert its output low-to-high level, and 25 ns for vice versa. The oscillation frequency ranges \sim MHz, so the peripheral components do not limit the TRNG operation as they operate at high speed.

We elaborated it in page 6 as below.

“Figure 2b shows 20 simulated oscillation waves obtained by the calculation, reproducing the experimental oscillation data in Fig. 1d very well. We assumed the device capacitance is 85 pF, and the parasitic capacitances are 42 pF. More detailed information regarding the circuit simulation can be found in **Supplementary Fig. S4.**”

We also described it in page 8 for more details as follows.

“The NbO_x TS device was loaded in the probe station and connected to the

breadboard via cables. In this setup, the parasitic capacitance in the breadboard is ~ 1 pF, and the op-amp and the flip-flop operate at a high speed of tens of ns. Thus, the signal delay by other components but the TS unit is negligible. More detailed discussions can be found in **Supplementary Fig. S4**. Figure 4c shows the monitored input and output signals for 5 random bit generation cycling.

#4. If an on-chip integrated solution is provided in the future, will the result of the paper (such as bit rate due to different parasitic, power consumption, area overhead) change?

(Author's response)

The speed bottleneck of our device is the TS unit itself. Others are transistor-based components and recent integration technology can achieve as high of GHz in speed. Thus, the speed bottleneck will be still the TS unit even in the integrated circuit.

As the review wrote in comment #3, the oscillation speed is related to the RC delay. In the integrated device, the device capacitance can be significantly reduced by scaling, resulting in the decrease of the RC delay. Also, the parasitic capacitances (C_{p1} and C_{p2}) due to wiring and measurement system can be eliminated. The resistance is almost identical as it is already very small. Overall, it is expected that the RC delay can be reduced in the integrated circuit and the oscillation frequency can be higher.

However, the accumulation of thermal fluctuation is mainly proportional to time, not the frequency. Thus, even the oscillation frequency increased, the bit generation rate may not be changed if the thermal environment is the same.

We added one paragraph at the end of conclusion to discuss it.

In page 12,

“Lastly, the accumulation of the thermal fluctuation is proportional to time but independent of the frequency. Thus, even at the integrated device having less parasitic components and oscillating at a higher frequency, the bit generation rate will not change as long as the operating temperature is the same. Therefore, embodying a heater near the TS unit can be a promising perspective of research for developing the higher speed of TRNG.”

#5. The NIST testing was collected with 25 μ s pulse (or 40kbs⁻¹), but in a different paragraph, the authors claim “This gives a random bit generation rate of 100 kbs⁻¹, which is the fastest record, compared with previous TS device-based TRNGs^{4,6,8}.”. What did not use the 100 kbs⁻¹ one for the NIST test?

(Author's response)

Thank you for careful reading our manuscript. Figure 4 demonstrated a specific case, the monobit test, which is the first rank of the NIST test. The bit generation rate of 100 kb/s is an example how we can define the TRNG speed from the generated bits. It can vary by testing type. Thus, to guarantee a successful random bit generation for all testing, we used 25 μ s pulses. Accordingly, the bit generation rate was 40 kb/s for all NIST randomness test.

We revised the corresponding paragraph not to confuse the readers in the page 9 as below,

“When the time was longer than $\sim 10 \mu\text{s}$, the P-value for monobit test exceeded 0.01 continuously, implying that the output is a random number. This gives a random bit generation rate of 100 kb/s for the monobit test. The random bit generation rate could be different by test types. For passing all 15 NIST randomness tests, we collected 130 sets of one Mbit (total 130 Mb) from the device using a 25 μs pulse (equivalent to 40 kb/s). Table 1 shows the testing results; The dataset passed all 15 NIST randomness tests successfully.”

#6. Could the authors also comment on the bit rate dependence on the temperature? Fig. 5 shows the oscillation at 390 K is quite different from that at 300 K (room temperature). Does it mean it is close to the upper temperature limit of the device? How about low temperature?

(Author's response)

Yes, as the review wrote, 390 K was the upper temperature limit of the device. With increasing the ambient temperature, both the threshold voltage (V_{th}) and hold voltage (V_{h}) decrease, and eventually, the NbO_x-based mott memristor loses its switching characteristics as the NDR diminishes at high temperature^{R5}.

We clarified it in page 10 as below.

“Figure 5a shows the current controlled NDR curves measured at different ambient temperatures ranging from 300 K to 390 K. Both the V_{th} and V_{h} decrease as the temperature increases, which is consistent with the precious reprot⁴². During the temperature testing, only the TS unit was heated in the probe station while other components were placed at room temperature. Above 390 K, the TS unit lost the NDR characteristic so that the oscillation was not observed. Thus, 390 K was the maximum operating temperature of our device. Figure 5b shows the oscillation waves at each temperature.”

As the TRNG speed is dependent on the ambient temperature as shown in Fig. 5c, the TRNG speed must be slower at the lower temperature as the thermal fluctuation is less active. We are sorry that we cannot provide the low temperature data and low temperature limitation value because it requires a specific setup in the measurement system. But, we are sure that the temperature dependent data in Fig. 5c is enough to show the temperature

dependence of our device.

- R5 Nandi, S. K., Li, S., Liu, X. & Elliman, R. G. Temperature dependent frequency tuning of NbOx relaxation oscillators. *Applied Physics Letters* **111**, 202901 (2017).

Reviewers' Comments:

Reviewer #1:

Remarks to the Author:

The authors have revised the manuscript according to the comments and suggestions from the reviewers. While it reads better than the first version, there are still some issues that should be addressed.

First of all, the novelty of this work, in my opinion, is the self-clocking scheme, in which the timing information for both the stimulus (the longer pulse) and the oscillating pulses is utilized. There is an obvious advantage without using an external counter. This main point should be better elaborated to stand out. The fundamental principle to generate dimensionless values is inspired by prior work (e.g., refs. 6 and 7). Other points, such as the 'long switching delay time' for prior arts are actually trivial because they can be engineered through pulse shape and materials stack.

Second, the method for the endurance measurements is debatable. The endurance was calculated by the oscillation frequency times the total time, as shown in Figure S3. However, the current level for the oscillating pulses is 0.3 mA, while that for a type NDR-2 (and hence a successful TRNG operation) is 0.6 mA (Figs. 1b, 1c). The discrepancy in the current level suggests that during the endurance measurements, the devices were not stressed to the level at which it operates as a TRNG. As such the claimed endurance data is questionable.

A side note on the current level is in Fig. 4c. The current level is much lower so that an operational amplifier is required for the circuits. Clarification on this low current level is needed as well.

Third, the estimation of energy consumption and comparison with other TRNGs are unfair. First of all, when using the data from the paper, I got a different result from the 5.23 nJ/bit. With a voltage of 1.45 V, the current is at 1 mA level, as such the energy required to generate a bit random number is $1.45V \times 0.9mA \times 20\mu s = 26$ nJ/bit. This does not even include the energy damped into the series resistor that is crucial to the oscillation behavior and the energy consumed by other electronic components such as the op-amp and the flip flop. Furthermore, it appears that the device is actually heated on a probe station during operation--- there is extra thermal energy in the authors' TRNG. All these should be taken into account.

Fourth, the authors discussed the variation tolerant testing results at different ambient temperatures and at different cells for their TRNG. However, Figs. 5c and 5f are merely based on 100 bits of data for each temperature or each cell, making the conclusions not convincing. The authors should collect many more random bits and run NIST tests again.

Finally, the authors should comment on how to further increase the bit generation rate. Based on the authors' estimation, the rate will be limited to around 66 kb/s at 380K. This is still low for high-speed applications, let alone the impracticality of "embody a heater near the TS unit".

Reviewer #2:

Remarks to the Author:

The authors have satisfactorily addressed my previous comments in this revised version. I have no further questions.

Reviewer #3:

Remarks to the Author:

The responses from the authors have cleared most concerns from the reviewer, and the reviewer agrees that the manuscript has improved after the revision.

The only concern that remains is about the endurance performance because as a random source, the reviewer does believe it requires a much higher endurance performance. The author further provides the endurance data (4×10^8) of the TRNG with the NbOx device in the revised manuscript, which is reasonably high. The authors mentioned that it is good for practical application. To justify this claim, it is recommended that the authors further provide a specific application scenario where 2.4×10^7 random bits generation is enough. Either way, the reviewer still believes the manuscript worth publication, as it shows what an emerging device is capable of after the performance issue is solved in the future, which itself is beyond the scope of this work.

Reviewer #1 (Remarks to the Author):

The authors have revised the manuscript according to the comments and suggestions from the reviewers. While it reads better than the first version, there are still some issues that should be addressed.

(Author's response)

Thank you for your valuable comments on our manuscript. We prepared one-by-one responses to the reviewer's comments carefully.

First of all, the novelty of this work, in my opinion, is the self-clocking scheme, in which the timing information for both the stimulus (the longer pulse) and the oscillating pulses is utilized. There is an obvious advantage without using an external counter. This main point should be better elaborated to stand out. The fundamental principle to generate dimensionless values is inspired by prior work (e.g., refs. 6 and 7). Other points, such as the 'long switching delay time' for prior arts are actually trivial because they can be engineered through pulse shape and materials stack.

(Author's response)

We highly agree that the self-clocking characteristic should be the main point of this study as we have already chosen 'self-clocking' as the first word of the title. We highlighted the importance of self-clocking capability more in the revised manuscript as below.

In page 3,

“In this study, we propose a self-clocking TRNG device utilizing the oscillating behavior of a mott memristor.”

In page 7,

“The origin of the stochastic oscillation is the random thermal fluctuation. To extract the randomness from it, we developed a novel TRNG circuit embodying a self-clocking capability. The self-clocking characteristic is the prominent feature of the proposed TRNG. It can generate the essential clock signal by itself, enabling the TRNG to be compact and energy-efficient.”

Second, the method for the endurance measurements is debatable. The endurance was calculated by the oscillation frequency times the total time, as shown in Figure S3. However, the current level for the oscillating pulses is 0.3 mA, while that for a type NDR-2

(and hence a successful TRNG operation) is 0.6 mA (Figs. 1b, 1c). The discrepancy in the current level suggests that during the endurance measurements, the devices were not stressed to the level at which it operates as a TRNG. As such the claimed endurance data is questionable.

(Author's response)

Thank you for the careful reading of our data. We understood the concerns by the reviewer, and we confirmed that the TS device during the endurance test was experienced the NDR-2. The current level discrepancy was originated from the different I – V characteristics of the device used for Fig 1b and Fig S3. Below Fig. S3(a) shows the I – V characteristic of the device used for the endurance testing in Fig. S3. We modified Fig. S3 as below.

Supplementary Figure 3 | Endurance characteristics of the NbOx-based mott device. For endurance testing, we applied a constant DC voltage to the NbOx-based oscillator and monitored the oscillating pulses per every 40 seconds with a load resistance of 4 k Ω . **a** shows the I – V curve of the device used for endurance test. **b** shows three pulse trains (reading – cycling – reading) whose widths are 20 μ s, 40 s, and 20 μ s, respectively. The pulse amplitude was 2.2 V. The reading pulse verifies whether the oscillator works or not. The cycling pulse operates the device for several million times approximately. The oscillation data at the reading period after **c** 120 s, **d** 240 s, **e** 480 s, and **f** 600 s are shown. The number of switching was calculated by the following equation; $f_{avg,s} \times t$, where $f_{avg,s}$ is the average oscillation frequency of 0.67 MHz, and t is 600 seconds. It gives 4×10^8 cycles. Our TRNG can generate one bit per 25 μ s. Thus, it can generate at least 2.4×10^7 random bits per device, which is reasonably high for the practical application.

The endurance-tested device shows NDR-2 between 0.2 mA to 0.23 mA, and the oscillation current peak reaches over 0.25mA, confirming the endurance test was performed under the stress over NDR-2.

The switching characteristics of the TS devices can be varied from cell to cell. The variation is mainly originated from the stochastic electroforming process even at the same electroforming conditions. Even though the NDR-2 current level is different, the internal temperature during the oscillation could be consistently higher than the mott transition temperature during oscillation.

A side note on the current level is in Fig. 4c. The current level is much lower so that an operational amplifier is required for the circuits. Clarification on this low current level is needed as well.

(Author's response)

The device in Fig. 4c has its NDR-2 region at around 0.2 mA and its oscillation peak current was lower. We clarified it as below.

In page 9,

“The input pulse amplitude and width were 1.1 V and 20 μ s, respectively, with 1 μ s of pulse rising and falling time. Although the oscillation peak current level was relatively low to 0.2 mA, it could provide sufficient voltage drop to the amplifier. The video recorded TRNG operation can be found in the **Supplementary Video.**”

Adding more discussion for clarification, in our TRNG, the oscillating current is not crucial. What matter is the oscillating voltage determined by the V_h and V_{th} of the TS device. During oscillation, the voltage potential on node-2 is oscillating from V_h to V_{th} regardless of the oscillation current and the input voltage. The op-amp amplifies the node-2 voltage as an input to a higher output voltage enough to trigger the flip-flop. In the revision, we clarify and elaborate the operating principles in detail as follows.

In page 7, we added one paragraph to elaborate the TRNG circuit operation as below.

“The origin of the stochastic oscillation is the random thermal fluctuation. To extract the randomness from it, we developed a novel TRNG circuit embodying a self-clocking capability. The self-clocking characteristic is the prominent feature of the proposed TRNG. It can provide the essential clock signal by itself, enabling the TRNG to be compact and energy-efficient as it allows to remove an external clock component. **Figure 3a** shows the TRNG circuit, including an input voltage source (V_{pulse}), a TS oscillator part (red square), a non-inverting amplifier part (blue square), and a negative edge triggered T flip-flop (green square). The 50 Ω of resistor represents the oscilloscope’s internal resistance. In this configuration, the TS oscillator converts the constant voltage input at node 1 to an oscillating voltage output at node 2. The oscillating voltage output approximately from a low value of V_h to a high value of V_{th} , where the V_{th} and V_h are the threshold and the hold voltages of the TS device, respectively. Also, the transition from high to low voltage is more drastic than the opposite transition because the heating rate is faster than the cooling rate¹⁸.”

Thus, we chose a negative edge triggered T flip-flop to utilize the drastic voltage drop as a clock signal. According to the T flip-flop's specification, its peak voltage should be higher than 1.25 V to be a valid clock signal. However, the V_{th} was lower than 1.25 V with some variation, meaning it could not feed a stable clock signal. The non-inverting amplifier can pull up the input voltage by $(1+R_f/R_s)$ times to 1.25 V of the required clock voltage, where the R_f and R_s values are adjustable considering the V_{th} . Then, the T flip-flop utilizes the amplified oscillation voltage at node 3 for the clock signal.”

Also, we revised Fig. 3 to show the TRNG operation more clearly. Accordingly, we revised the corresponding part in the manuscript as below.

Fig. 3 is revised as below.

Figure 3 | A TRNG circuit and its simulation. **a** The proposed TRNG circuit composed of an input voltage source (V_{pulse}), a TS oscillator part (red square), a non-inverting amplifier part (blue square), and a negative edge triggered T flip-flop (green square). **b** The input voltage pulse (black) at node 1 and the output voltage (red) and current (bright blue) at node 2. **c** The amplified oscillation voltage at node 3 (blue) compared with the output voltage (red) and current (bright blue) at node 2. The horizontal line at 1.25 V indicates the required clock's peak voltage for T flip-flop operation. **d** The output current at node 4 (green) with respect to the amplified oscillation voltage as a clock of the T flip-flop. **e** The input voltage at node 1, the amplified voltage at node 3, and the output current at node 4 during the entire time frame for generating one random bit. In this case, the random output is 1.

The discussion part of Fig. 3 in page 8 is revised as below.

“**Figure 3b** shows the input voltage pulse at node 1 and the stochastic oscillation

current and voltage at node 2 obtained by experiments. **Figures 3c-e** show the monitoring signals at nodes 2-4 in **Fig. 3a** obtained by a PSPICE simulation for validating the circuit operation. **Figure 3c** displays the oscillation current and voltage at node 2 and the amplified voltage at node 3. The horizontal line at 1.25 V indicates the required clock's peak voltage for T flip-flop operation. The op-amp amplifies the input voltage to trigger the subsequent flip-flop properly. If the oscillation voltage can be increased, the op-amp can be removed, and the device architecture can be more compact. **Figure 3d** shows the output current at node 4 with the clocking voltage at node 3.

Third, the estimation of energy consumption and comparison with other TRNGs are unfair. First of all, when using the data from the paper, I got a different result from the 5.23 nJ/bit. With a voltage of 1.45 V, the current is at 1 mA level, as such the energy required to generate a bit random number is $1.45\text{V} \cdot 0.9\text{mA} \cdot 20\mu\text{s} = 26 \text{ nJ/bit}$. This does not even include the energy damped into the series resistor that is crucial to the oscillation behavior and the energy consumed by other electronic components such as the op-amp and the flip flop. Furthermore, it appears that the device is actually heated on a probe station during operation-- there is extra thermal energy in the authors' TRNG. All these should be taken into account.

(Author's response)

We are sorry for the confusion from missing of detailed explanation on the energy calculation. In our calculation, we used an average current ($I_{\text{AVG}} = 144 \mu\text{A}$) instead of the peak current ($I_{\text{MAX}} = 0.9 \text{ mA}$) during oscillation. The average current was calculated from the raw oscillation data by integrating the current over time and dividing the total current by time. By that way, we could obtain 5.23 nJ/bit.

This calculated energy consumption includes energy dissipation at the series load resistor as we multiplied the input voltage ($V_A = 1.45 \text{ V}$) to the entire circuit, not the dynamic node voltage on the TS device. Thus, it has already considered the energy damped into the series resistor.

The total energy consumption of the TRNG device is strongly rely on the active components, not the TS device because the active components typically consume about several hundreds of mW. Then, the energy consumption at the active components is proportional to time, thus, a faster bit generation is crucial for reducing the total energy consumption. Below table compares the estimated energy consumption at the TS unit and the required active components of various volatile-memristor-based TRNGs. In this work, not only the clock generator was not used, but also the number of other active components is less than other studies^{R1, R2}.

We added Fig. S3 in the supplementary information to compare energy consumption of volatile memristor-based TRNGs as below.

In Fig. S3,

Specification comparison table of volatile-memristor-based TRNGs								
	Bit generation rate ($\mu\text{s}/\text{bit}$)	TS memristor			Clock generator		Other active components	
		Power		Energy consumption (nJ/bit)	Clock generator power	Energy consumption (nJ/bit)	Total number (List)	Energy consumption (nJ/bit)
		Operating voltage (V)	Operating current (mA)					
Jiang, H. et al. [1]	166.6	0.4	10^{-5}	0.67×10^{-3}	~ 150 mW	$\sim 2.5 \times 10^4$	4 (Comparator, AND gate, Counter $\times 2$)	Each active component consumes \sim mW.
Woo, K. S. et al. [2]	62.5	10	10^{-6}	0.63×10^{-3}	~ 150 mW	$\sim 9.4 \times 10^3$	6 (XNOR, XOR gates, D flip-flop $\times 4$)	
This work	25	1.45	0.144 (in average during oscillation)	5.22 (including the clock)	-	0	2 (Op-amp, T flip-flop)	

Figure S3 | The energy consumption comparison table between volatile-memristor based TRNGs. The clock generator is the highest energy-consuming part. A typical ultra-low-power clock generator, CDCI6214 by Texas Instruments, consumes 150 mW of power. The clock generator should be active during the entire bit generation time. Thus, they consume about 25 $\mu\text{J}/\text{bit}$ in ref. 1 and 9.4 $\mu\text{J}/\text{bit}$ in ref. 2, which is far higher than the energy consumption in the TS memristor. Our TRNG does not require the external clock generator due to its inherent self-clocking characteristic, allowing a significant energy consumption reduction. Moreover, the number of other active components is also the minimum, confirming our TRNG is the most compact and energy-efficient.

References

- 1 Jiang, H. *et al.* A novel true random number generator based on a stochastic diffusive memristor. *Nat Commun* **8**, 882 (2017).
- 2 Woo, K. S. *et al.* A Combination of a Volatile - Memristor - Based True Random - Number Generator and a Nonlinear - Feedback Shift Register for High - Speed Encryption. *Advanced Electronic Materials* **6**, 1901117 (2020).”

In addition, we also revised the manuscript in page 8 as below.

“The self-clocking ability can be advantageous in energy as well. The energy consumption of the TS oscillator in Fig. 1c was 5.23 nJ/bit ($V_A = 1.45$ V, $I_{\text{avg}} = 144$ μA , $t_{\text{TRNG}} = 25$ $\mu\text{s}/\text{bit}$), where V_A is the applied operating voltage, I_{avg} is the average current during operating, and t_{TRNG} is the required time for true number generation. The I_{avg} is obtained by integrating the current during oscillation and dividing it by t_{TRNG} . It is higher than energy consumption of another TS memristors in previous TRNGs of 0.67 pJ/bit ($V_A = 0.4$ V, $I_{\text{avg}} = 10^{-8}$ A, $t = 166.6$ $\mu\text{s}/\text{bit}$)⁶ and 0.63 pJ/bit ($V = 10$ V, $I = 10^{-9}$ A, $t = 62.5$ $\mu\text{s}/\text{bit}$)⁸. However, they required an external clock generator that consumes about several hundred mW, which is much higher than the TS oscillator. (The power consumption of CDCI6214 by Texas Instruments, ultra-low-power clock generator, is 150 mW). Moreover, in our TRNG, less active components are

used so that the total energy consumption can also be more efficient. More detailed energy comparison can be found in **Supplementary Fig. S3.**”

The heating experiments was limited to Fig. 5 for checking the environmental temperature-tolerance of our TRNG. Therefore, it is not the energy-consumption at the TRNG device.

Fourth, the authors discussed the variation tolerant testing results at different ambient temperatures and at different cells for their TRNG. However, Figs. 5c and 5f are merely based on 100 bits of data for each temperature or each cell, making the conclusions not convincing. The authors should collect many more random bits and run NIST tests again.

(Author's response)

In Figs. 5c and 5f, we examined only the monobit test, which is the mandatory and the first priority test among the NIST randomness test. According to the monobit test instruction at the guideline of NIST special publication 800-22 test, the recommendation size of input data is more or equal to 100 bits, and our data size fulfills it. We copied the corresponding part of the instruction as below for reference.

In monobit test instruction, [Rukhin A., S. J., Nechvatal J., Smid M., Barker E., Leigh S., Levenson M., Vangel M., Banks D., Heckert A., Dray J., Vo S. NIST Special Publication 800-22. (2010)],

“It is recommended that each sequence to be tested consist of a minimum of 100 bits (i.e., $n \geq 100$).”

We clarified it in page 10 in our manuscript as follows.

“The monobit test is the first test among the 15 tests in NIST (National Institute of Standards and Technology) random number test suite (NIST 800-22)⁴¹ checking whether the fraction of ones in the given random bits is close to 1/2 or not, and it is the essential test because other tests would fail if the monobit test fails⁴¹. The recommended minimum data size for monobit test is 100 bits according to the instruction. The P-value is an indicator showing the randomness of the dataset; when the value is higher, the dataset is more random. In general, if the P-value exceeds 0.01, the dataset can be regarded as true random numbers.”

Please understand that collecting millions of data needs excessive time and efforts for setting-up and the output data treatment. We can do it if it is mandatory for publication. However, we believe the rationality of our argument is enough as is.

Finally, the authors should comment on how to further increase the bit generation rate. Based on the authors' estimation, the rate will be limited to around 66 kb/s at 380K. This

is still low for high-speed applications, let alone the impracticality of "embody a heater near the TS unit".

(Author's response)

In our TRNG, it utilized a periodic but stochastic oscillation of the TS device. Within the framework, there could be an inherent limitation in the bit generation rate as the reviewer correctly understood. Embodying the heater near the TS device is not just a conceptual idea. We added a reference that demonstrated it. [del Valle, J., Salev, P., Kalcheim, Y. & Schuller, I. K. A caloritronics-based Mott neuristor. *Scientific Reports* **10**, (2020)]

To further increase the bit generation rate, the oscillation should be chaotic. Some studies have reported the chaotic oscillation characteristic from the NbO_x based mott memristor.[Kumar, S., Strachan, J. P. & Williams, R. S. Chaotic dynamics in nanoscale NbO₂ Mott memristors for analogue computing. *Nature* **548**, 318-321 (2017), and Kumar, S., Williams, R. S. & Wang, Z. Third-order nanocircuit elements for neuromorphic engineering. *Nature* **585**, 518-523 (2020)] Unfortunately, we could not find such a chaotic condition yet in our device. If the chaotic oscillation is possible, one can generate one random bit per every oscillation, and the bit generation rate can be as high as ~Mbit/s or higher.

We added the discussion at the end of the conclusion part in page 13 as below.

“Lastly, the accumulation of the thermal fluctuation is proportional to time but independent of the frequency. Thus, even at the integrated device having fewer parasitic components and oscillating at a higher frequency, the bit generation rate will not be changed as long as the operating temperature is the same. For developing the higher speed of TRNG, embodying a heater near the TS unit can be plausible⁴⁴. Also, some studies reported a chaotic oscillation behavior from the TS device at a controlled condition^{16,17}. If such chaotic oscillation can be offered reliably, the bit generation rate could be as high as ~Mbit/s, equal to the oscillation frequency. These can be a promising perspective of research in the next.

”

—

Reviewer #3 (Remarks to the Author):

The responses from the authors have cleared most concerns from the reviewer, and the reviewer agrees that the manuscript has improved after the revision.

The only concern that remains is about the endurance performance because as a random source, the reviewer does believe it requires a much higher endurance performance. The author further provides the endurance data (4×10^8) of the TRNG with the NbOx device in the revised manuscript, which is reasonably high. The authors mentioned that it is good for practical application. To justify this claim, it is recommended that the authors further provide a specific application scenario where 2.4×10^7 random bits generation is enough. Either way, the reviewer still believes the manuscript worth publication, as it shows what an emerging device is capable of after the performance issue is solved in the future, which itself is beyond the scope of this work.

(Author's response)

One of the potential applications can be found at cryptography. For example, a public-key cryptosystem, Rivest-Shamir-Adleman (RSA)-2048 is widely used for secure data transmission (e.g. internet banking services) in most countries.[Bernstein, D. J., Lange, T. & Schwabe, P. The security impact of a new cryptographic library. in Lecture Notes in Computer Science (including subseries Lecture Notes in Artificial Intelligence and Lecture Notes in Bioinformatics) 7533 LNCS, 159–176 (Springer Verlag, 2012).] The key size of RSA-2048 is 2048 bits, so our TRNG can generate $\sim 10^4$ keys for data encryption. For one example, a person can use a secured internet banking for $\sim 10^4$ times, which is sufficient.

We elaborated our manuscript in page 10 as follows.

“The endurance of our TRNG was about 4×10^8 cycles. Thus, it can generate at least 2.4×10^7 random bits per device. It is reasonably high for some practical applications such as security code generation in cryptography that require random number generation by request, not continuously⁴².”

Reviewers' Comments:

Reviewer #1:

Remarks to the Author:

The authors have satisfactorily addressed my comments in the previous reviews.

Reviewer #3:

Remarks to the Author:

The authors have addressed my concerns and I am happy to support its publications.